# TET knockout cells transit between pluripotent states and exhibit precocious germline entry

Raphaël Pantier [1,2,3,5], Elisa Barbieri[1,2,5], Sara Gonzalez Brito [1,2,5], Ella Thomson [1,2], Tülin Tatar[1,2], Douglas Colby[1,2], Man Zhang[1,2,4] & Ian Chambers [1,2✉]

## Abstract

**TET1, TET2 and TET3 are DNA demethylases with important roles in development and differentiation. To assess the contributions of TET proteins to cell function during early development, single and compound knockouts of *Tet* genes in mouse pluripotent embryonic stem cells (ESCs) were generated. Here, we show that TET proteins are not required to transit between naïve, formative and primed pluripotency states. Moreover, ESCs with double knockouts of *Tet1* and *Tet2* or triple knockouts of *Tet1, Tet2* and *Tet3* are phenotypically indistinguishable. TET1,2,3-deficient ESCs exhibit differentiation defects and fail to activate somatic gene expression, retaining expression of pluripotency transcription factors. Therefore, TET1 and TET2, but not TET3 act redundantly to facilitate somatic differentiation. Importantly however, TET-deficient ESCs can differentiate into primordial germ cell-like cells (PGCLCs), and do so at high efficiency in the presence or absence of PGC-promoting cytokines. Moreover, acquisition of a PGCLC transcriptional programme occurs more rapidly in TET-deficient cells. These results establish that TET proteins act at the juncture between somatic and germline fates: without TET proteins, epiblast cell differentiation defaults to the germline.**

**Keywords** Primordial Germ Cells; TET Proteins; Pluripotency; Stem Cells; Differentiation
**Subject Categories** Chromatin, Transcription & Genomics; Development; Stem Cells & Regenerative Medicine

## Introduction

Ten eleven translocation (TET) proteins promote active DNA demethylation by catalysing the oxidation of 5-methylcytosine (Tahiliani et al, 2009; Ito et al, 2011; He et al, 2011). In mammals, TET proteins are encoded by three genes (*Tet1, Tet2* and *Tet3*) which share an evolutionary conserved catalytic domain. TET proteins interact with multiple transcription factors (Costa et al, 2013; Okashita et al, 2014; Rampal et al, 2014; Wang et al, 2015; Sardina et al, 2018; Pantier et al, 2020), which could mediate their targeting to enhancer elements (Lu et al, 2014; Hon et al, 2014; Bogdanović et al, 2016; Ginno et al, 2020; Charlton et al, 2020). Additionally, TETs interact with proteins modulating chromatin, such as *O*-glycosyltransferase (Vella et al, 2013; Deplus et al, 2013; Shi et al, 2013; Chen et al, 2013) and the SIN3A co-repressor complex (Williams et al, 2011; Chandru et al, 2018; Zhu et al, 2018). Such partner protein interactions may be important in mediating the critical role of TET proteins during embryonic development.

TET proteins are expressed during early embryonic development and in ESCs (Ito et al, 2010; Gu et al, 2011; Dawlaty et al, 2011; Koh et al, 2011). In mice, deletion of individual *Tet* genes is compatible with normal embryonic development but does lead to tissue-specific phenotypes during later foetal development and in adults (Dawlaty et al, 2011; Li et al, 2011; Gu et al, 2011; Yamaguchi et al, 2012). In particular, although initial germline commitment occurs normally, TET1 deficiency leads to defects during later gametogenesis, including incomplete reprogramming of genomic imprints and defective oocyte meiosis (Yamaguchi et al, 2012, 2013; Hackett et al, 2013; SanMiguel et al, 2018; Hill et al, 2018). Loss of TET2 causes defects in hematopoietic stem cell differentiation, leading predominantly to myeloid malignancies in adults (Li et al, 2011; Quivoron et al, 2011; Moran-Crusio et al, 2011; Ko et al, 2011; Muto et al, 2014). TET3 loss causes neonatal lethality, potentially due to its function in the reprogramming of the paternal genome in fertilised zygotes (Gu et al, 2011; Wang et al, 2013; Tsukada et al, 2015). In addition, the combined loss of TET1 and either TET2 or TET3 can cause morphological abnormalities and growth defects from embryonic day E10.5 (Dawlaty et al, 2013; Kang et al, 2015). Of note, a minority of *Tet1/2* double knockout mice can survive until adulthood and remain fertile (Dawlaty et al, 2013).

In contrast, embryos with triple knockouts of *Tet1, Tet2* and *Tet3* die shortly after implantation with gastrulation defects (Dai et al, 2016; Li et al, 2016). Consistent with this in vivo phenotype, ESCs with combined deletions of *Tet1, Tet2* and *Tet3* are blocked in somatic differentiation, whereas ESCs with single knockouts of *Tet*

[1]Centre for Regenerative Medicine, Institute for Regeneration and Repair, 5 Little France Drive, Edinburgh EH16 4UU, Scotland. [2]Institute for Stem Cell Research, School of Biological Sciences, University of Edinburgh, 5 Little France Drive, Edinburgh EH16 4UU, Scotland. [3]Present address: Institut de Génétique et de Biologie Moléculaire et Cellulaire (IGBMC), CNRS UMR7104, INSERM U1258, Université de Strasbourg, 1 rue Laurent Fries, Illkirch, Cedex, France. [4]Present address: Guangzhou National Laboratory, No. 9 XingDaoHuanBei Road, 510005 Guangzhou, China. [5]These authors contributed equally: Raphaël Pantier, Elisa Barbieri, Sara Gonzalez Brito. ✉E-mail: ichambers@ed.ac.uk

genes differentiate normally (Dawlaty et al, 2014; Verma et al, 2018). These studies indicate functional redundancy amongst TET proteins during early development of somatic cells. However, the abilities of cells lacking TET proteins to initiate differentiation into the germline have not been assessed. Here, we removed all *Tet* family genes, singly and in combination, and performed in vitro

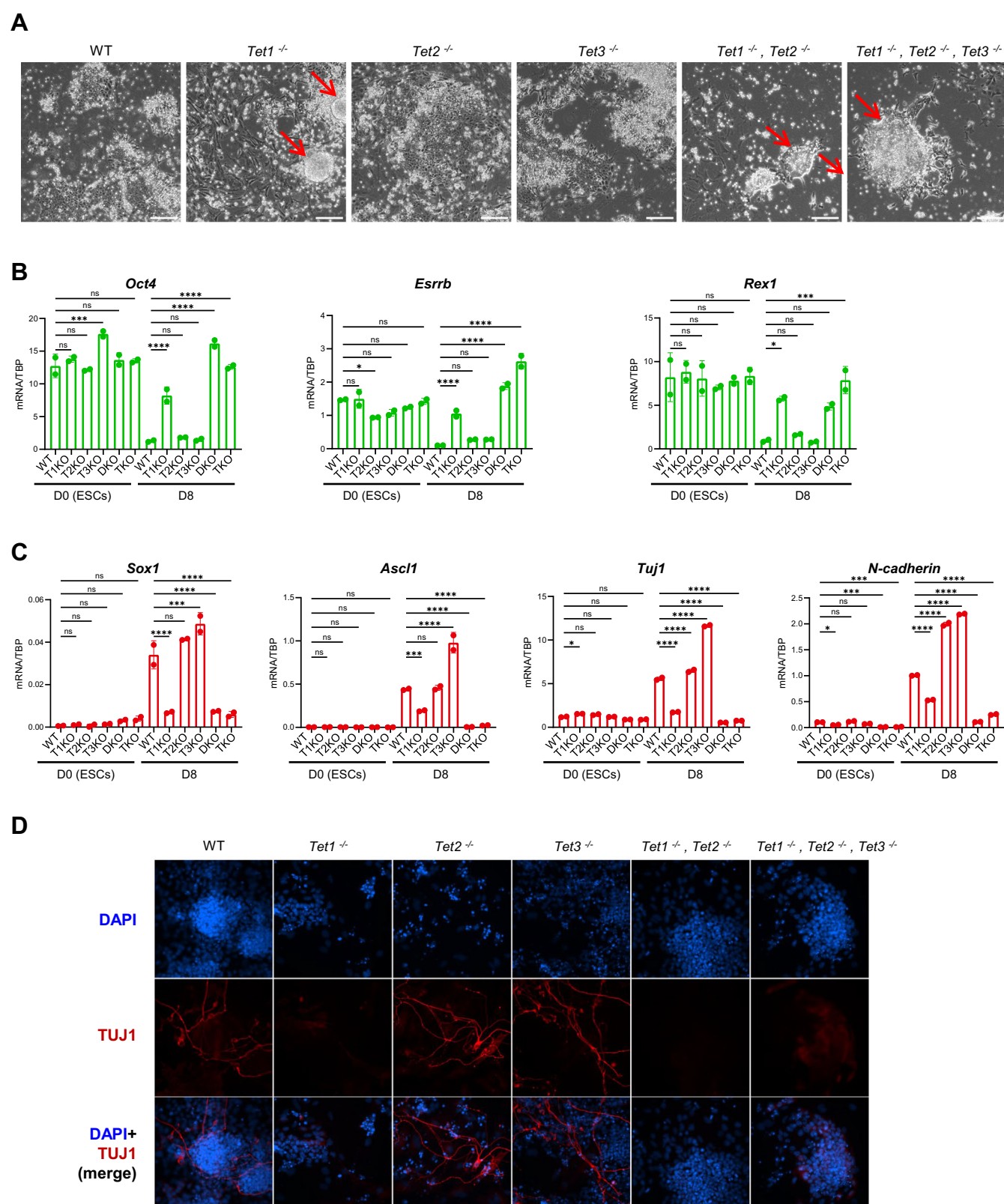

◄ **Figure 1. Phenotype of TET-deficient cells during monolayer neural differentiation.**

(A) Phase contrast images of the indicated live cell lines after monolayer neural differentiation for 6 days. Colonies retaining an undifferentiated morphology are indicated by red arrows. Scale bars: 100 μm. (B, C) mRNA levels of pluripotency factors (B) and neural markers (C) in the indicated cell lines cultured as ESCs in serum/LIF (D0) or after 8 days of monolayer neural differentiation (D8). mRNA levels were quantified by RT-qPCR and normalised to TBP mRNA levels. Data are from a representative of two independent experiments (data points: technical replicates, centre: mean, error bars: standard deviation, $n = 2$). Stars indicate statistical significance compared to wild-type (one-way ANOVA test). Individual $p$ values are provided in Table EV1. (D) Immunofluorescence for TUJ1 in the indicated cell lines after 8 days of monolayer neural differentiation ($n = 3$). Scale bar: 50 μm.

differentiation assays to monitor the relative contributions of TET1, TET2 and TET3 to cellular transitions during peri-implantation development. Comparative analyses of cells with single, double and triple *Tet* gene deletions confirmed the somatic differentiation block in TET-deficient cells. Our results extend previous findings by showing that the somatic differentiation block is not due to an inability of TET-deficient cells to transit from naïve, through formative to primed pluripotency. Our results also establish that TET-deficient cells are able to differentiate into the germline at high efficiency and do so at an accelerated rate and without the usual requirement for cytokine addition.

## Results

### Development of a novel strategy to knockout *Tet1/2/3* alleles by CRISPR/Cas9

Previous studies used a variety of approaches to knockout *Tet* alleles, including coding exon disruption by classic gene targeting (Li et al, 2011; Dawlaty et al, 2011; Gu et al, 2011; Yamaguchi et al, 2012; Zhang et al, 2013; Hu et al, 2014; Dai et al, 2016) or frameshift mutagenesis using CRISPR/Cas9 (Wang et al, 2013; Lu et al, 2014; Verma et al, 2018; Ginno et al, 2020). However, these strategies left open the formal possibility of truncated TET protein fragment expression. Therefore, to eliminate all endogenous *Tet1*, *Tet2* and *Tet3* coding potential, the *Tet* open reading frames were excised using CRISPR/Cas9 and two sgRNAs targeting the start and stop codons, respectively (Fig. EV1A–F). Single and combined *Tet* knockout cell lines were generated from wild-type E14Tg2a ESCs using a sequential strategy (Fig. EV1G). Since our previous analyses indicated that TET3 protein is undetectable in ESCs (Pantier et al, 2019), we focused our analysis of compound mutants on a comparison of *Tet1⁻ᐟ⁻*, *Tet2⁻ᐟ⁻* double knockout (DKO) and *Tet1⁻ᐟ⁻*, *Tet2⁻ᐟ⁻*, *Tet3⁻ᐟ⁻* triple knockout (TKO) cells. Targeted ESC lines were first assessed for expression of *Tet* mRNAs (Fig. EV1H). As expected, expression of *Tet1*, *Tet2* and *Tet3* mRNAs was lost in ESCs carrying deletions of *Tet1*, *Tet2* and *Tet3*, respectively. For most subsequent experiments, we compared single and combined *Tet* knockout cell lines side by side using one clone for each genotype (T1KO C3, T2KO C26, T3KO C3, DKO C13 and TKO C15, see Fig. EV1G).

### TET1 and TET2, but not TET3, are required for somatic lineage commitment

To assess the ability of *Tet* knockout ESCs to undergo somatic differentiation, two distinct protocols were used: monolayer neural differentiation and embryoid body formation. Following monolayer neural differentiation (Ying et al, 2003), flattened differentiated

cells appeared in wild-type, *Tet2⁻ᐟ⁻* and *Tet3⁻ᐟ⁻* cultures. In contrast, colonies with compact, rounded morphologies resembling ESC colonies persisted in *Tet1⁻ᐟ⁻* single knockout, *Tet1⁻ᐟ⁻*, *Tet2⁻ᐟ⁻* DKO and *Tet1⁻ᐟ⁻*, *Tet2⁻ᐟ⁻*, *Tet3⁻ᐟ⁻* TKO cultures (Fig. 1A). After 8 days of differentiation, *Tet1⁻ᐟ⁻* single knockout, *Tet1⁻ᐟ⁻*, *Tet2⁻ᐟ⁻* DKO and *Tet1⁻ᐟ⁻*, *Tet2⁻ᐟ⁻*, *Tet3⁻ᐟ⁻* TKO cultures also retained mRNAs encoding the pluripotency transcripts *Oct4*, *Esrrb* and *Rex1* (Fig. 1B). Compared to wild-type, *Tet2⁻ᐟ⁻* and *Tet3⁻ᐟ⁻* cell lines, *Tet1⁻ᐟ⁻* cells showed reduced induction of the neural markers *Sox1*, *Ascl1*, *Tuj1* and *N-cadherin (Cdh2)* (Fig. 1C). *Ascl1*, *Tuj1* and *N-cadherin (Cdh2)* mRNAs were reduced further in *Tet1⁻ᐟ⁻*, *Tet2⁻ᐟ⁻* DKO and *Tet1⁻ᐟ⁻*, *Tet2⁻ᐟ⁻*, *Tet3⁻ᐟ⁻* TKO cells (Fig. 1C). Consistent with these changes TUJ1-positive neurons were undetectable in *Tet1⁻ᐟ⁻* single knockout, *Tet1⁻ᐟ⁻*, *Tet2⁻ᐟ⁻* DKO and *Tet1⁻ᐟ⁻*, *Tet2⁻ᐟ⁻*, *Tet3⁻ᐟ⁻* TKO cultures (Fig. 1D). These results show that the functions of TET1 and, to a lesser extent, TET2 are required in combination for efficient neural differentiation.

Multi-lineage differentiation was assessed after embryoid body (EB) formation. This showed the persistence of some colonies with an undifferentiated ESC-like morphology for *Tet1⁻ᐟ⁻*, *Tet2⁻ᐟ⁻* DKO and *Tet1⁻ᐟ⁻*, *Tet2⁻ᐟ⁻*, *Tet3⁻ᐟ⁻* TKO cells (Fig. 2A). In contrast, wild-type or single knockout *Tet* cultures appeared fully differentiated (Fig. 2A). In addition, after 14 days *Tet1⁻ᐟ⁻*, *Tet2⁻ᐟ⁻* DKO and *Tet1⁻ᐟ⁻*, *Tet2⁻ᐟ⁻*, *Tet3⁻ᐟ⁻* TKO EBs retained expression of the pluripotency transcription factors *Oct4*, *Esrrb* and *Nanog* (Fig. 2B) and failed to upregulate the germ layer markers *Gata4*, *Col1a1*, *Flk1* and *Sox17* (Fig. 2C). Furthermore, *Tet1⁻ᐟ⁻*, *Tet2⁻ᐟ⁻* DKO and *Tet1⁻ᐟ⁻*, *Tet2⁻ᐟ⁻*, *Tet3⁻ᐟ⁻* TKO EBs also showed an almost complete lack of beating colonies, indicating a block to mature cardiomyocyte differentiation (Fig. 2D).

Together, these results extend previous analysis (Dawlaty et al, 2014; Verma et al, 2018) by showing that it is the combined actions of TET1 and TET2, and not TET3, that are required for multi-lineage specification.

### The combined loss of TET1 and TET2 enhances ESC self-renewal

To examine the effect of *Tet* gene deletions on the undifferentiated ESC phenotype, we assessed the morphology, self-renewal efficiency and expression of pluripotency markers of *Tet* knockout ESC lines. At routine passaging density, phase contrast imaging of live ESC cultures indicated that loss of TET proteins did not affect ESC morphology (Fig. EV2A). To quantitatively assess self-renewal efficiency, *Tet* knockout ESCs were plated at clonal density, cultured for 7 days in the presence or absence of leukaemia inhibitory factor (LIF) and stained for alkaline phosphatase activity (Figs. 3A and EV2B). Interestingly, both *Tet1⁻ᐟ⁻*, *Tet2⁻ᐟ⁻* DKO and *Tet1⁻ᐟ⁻*, *Tet2⁻ᐟ⁻*, *Tet3⁻ᐟ⁻* TKO ESCs showed a strong increase in the proportion of undifferentiated colonies formed in the presence of

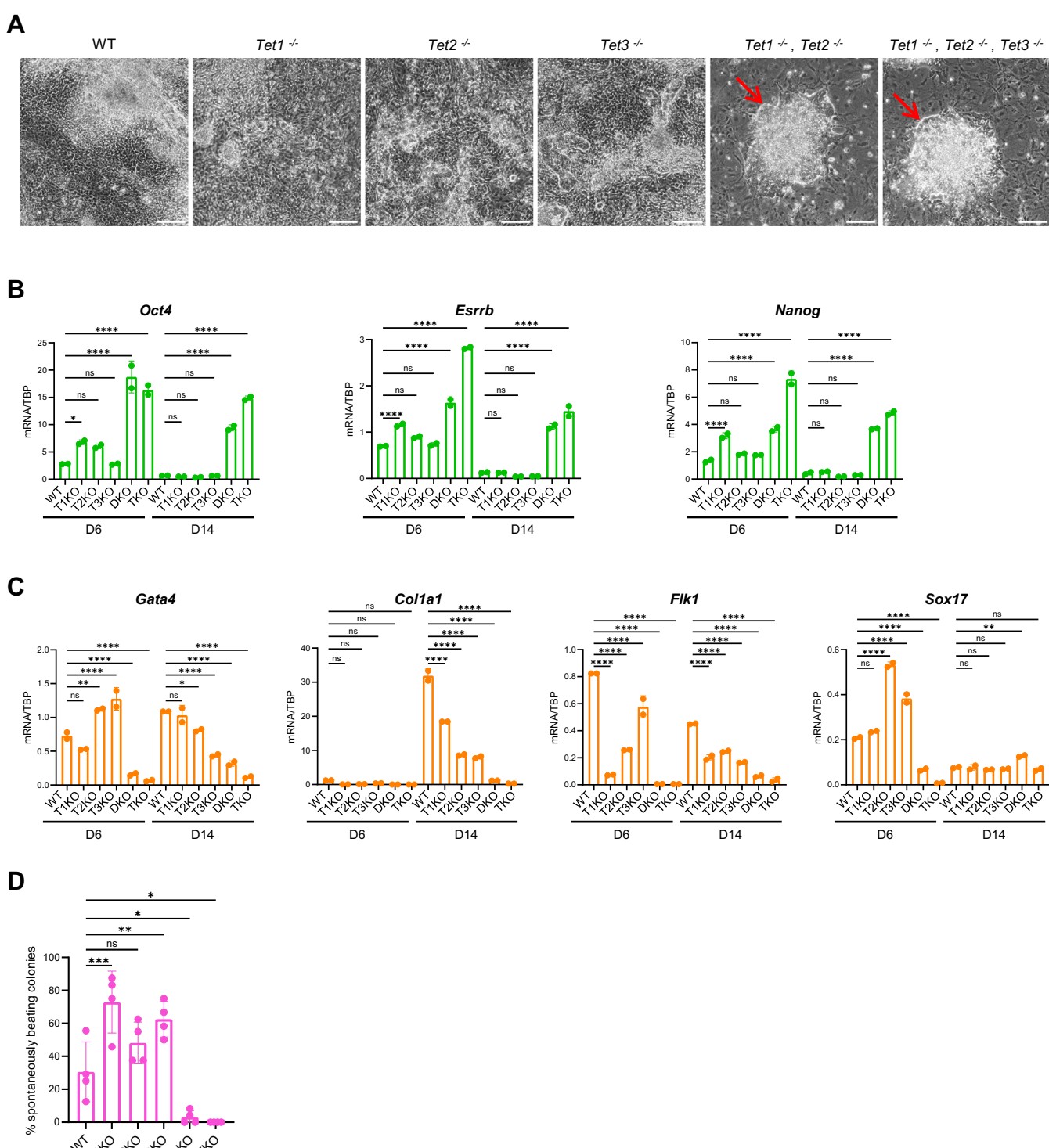

**Figure 2. Phenotype of TET-deficient cells during embryoid body differentiation.**

(A) Phase contrast images of the indicated live cell lines following 13 days of embryoid bodies (EB) differentiation. Colonies retaining an undifferentiated morphology are indicated by red arrows. Scale bars: 100 μm. (B, C) mRNA levels of pluripotency factors (B) and lineage markers (C) in the indicated cell lines following embryoid body formation (D6) and after a further 8 days of differentiation as outgrowths (D14). mRNA levels were quantified by RT-qPCR and normalised to TBP mRNA levels. Data are from a representative of two independent experiments (data points: technical replicates, centre: mean, error bars: standard deviation, $n = 2$). Stars indicate statistical significance compared to wild-type (one-way ANOVA test). Individual p values are provided in Table EV1. (D) Proportion (%) of spontaneously beating colonies following 13 days of EB differentiation with the indicated cell lines to assess cardiomyocyte differentiation (data points: replicate experiments, centre: mean, error bars: standard deviation, $n = 4$). Stars indicate statistical significance compared to wild-type (one-way ANOVA test). Individual p values are provided in Table EV1.

LIF (Fig. 3A). However, no differences were seen in the colonies formed by any of these lines in the absence of LIF (Fig. EV2B). This increased responsiveness to LIF was reflected in the morphology of individual colonies formed by *Tet1*⁻/⁻, *Tet2*⁻/⁻ DKO and *Tet1*⁻/⁻, *Tet2*⁻/⁻, *Tet3*⁻/⁻ TKO ESC clones, which were more undifferentiated than wild-type or single *Tet* knockout ESCs, with fewer differentiated cells present at the edges (Fig. 3B). Together these results indicate that the redundant activities of TET1 and TET2 reduce the efficiency of ESC self-renewal.

However, co-immunofluorescence analysis indicated that single and combined *Tet* gene deletions did not affect the levels and distribution of the pluripotency factors NANOG and OCT4 in cell cultures (Fig. 3C). Notably, NANOG remains heterogeneously expressed in *Tet* mutant lines, suggesting that loss of TET proteins may not affect transitions between the pluripotent cell states present in serum-containing cultures (Chambers et al, 2007).

## Loss of TETs does not affect transitions from naïve to formative or primed pluripotency

To further assess the role of TET proteins during the transitions between pluripotent states, *Tet* knockout ESCs were differentiated into epiblast-like cells (EpiLCs) (Hayashi et al, 2011; Hayashi and Saitou, 2013). All *Tet* mutant cell lines differentiated into EpiLCs within 48 h, without noticeable morphological differences between wild-type and *Tet* knockout lines (Fig. EV2C). To assess transcriptional changes accompanying the transition from naïve to formative pluripotency, RNA was analysed at 0 and 48 h of EpiLC differentiation. In *Tet* knockout and wild-type cells, the levels of the naïve pluripotency mRNAs *Esrrb*, *Prdm14* and *Rex1* rapidly decreased, while *Oct4* mRNA remained expressed (Fig. 4A). Conversely, the formative pluripotency markers *Otx2* and *Oct6* were upregulated in a similar manner in wild-type and *Tet* knockout EpiLCs (Fig. 4B). These results indicate that TET proteins are dispensable for the transition from naïve to formative pluripotency.

To assess the requirement for TET protein function for the transition to primed pluripotency, *Tet* knockout ESCs were differentiated into epiblast stem cells (EpiSCs) in vitro (Guo et al, 2009). Stable self-renewing EpiSC lines lacking either single or combined *Tet* alleles were obtained. Phase contrast microscopy indicated that all mutant lines had a typical flat, elongated morphology (Fig. EV2D). Co-immunofluorescence analysis confirmed that the primed pluripotency factors OCT6 and SOX2 were expressed in all *Tet* knockout EpiSC lines, similarly to wild-type cells (Fig. 4C). As EpiSCs resemble epiblast cells in early post-implantation embryos at ~E5.5-6.5 (Tesar et al, 2007; Brons et al, 2007; Han et al, 2010), these results are compatible with the fact that *Tet1*⁻/⁻, *Tet2*⁻/⁻, *Tet3*⁻/⁻ triple knockout embryos are undistinguishable from wild-type at this developmental stage (Dai et al, 2016; Li et al, 2016).

Together, our results extend previous reports that *Tet1*⁻/⁻, *Tet2*⁻/⁻, *Tet3*⁻/⁻ triple knockout cells are impaired in somatic differentiation in vitro (Dawlaty et al, 2014; Li et al, 2016; Verma et al, 2018) by showing that this is also the case for *Tet1*⁻/⁻, *Tet2*⁻/⁻ double knockout cells and by showing that *Tet* mutants are unimpaired in their ability to transit between pluripotent states.

## TET-deficient cells efficiently acquire germline markers without requiring inductive cytokines

To assess the ability of *Tet* mutant cells to enter germline development, wild-type and *Tet* knockout ESCs were transitioned

to an EpiLC state before further differentiation into primordial germ cell-like cells (PGCLCs) (Hayashi et al, 2011; Hayashi and Saitou, 2013). After 6 days, wild-type and single *Tet* deleted lines produced similar low levels of cells displaying the PGCLC surface markers SSEA-1 and CD61 (Fig. 5A,C). In contrast, the proportion of SSEA1⁺/CD61⁺ cells was strongly increased in both *Tet1*⁻/⁻, *Tet2*⁻/⁻ DKO and *Tet1*⁻/⁻, *Tet2*⁻/⁻, *Tet3*⁻/⁻ TKO lines (Fig. 5A,C). This result is reminiscent of the increased proportion of SSEA1⁺/CD61⁺ cells formed in cells deleted for the transcription factor *Otx2* (Zhang et al, 2018a). In the case of *Otx2*-null cells, ~30% of cells could even form SSEA1⁺/CD61⁺ cells without the usual requirement for PGC-promoting cytokines. To determine if this was also the case for *Tet* mutants, differentiation was performed in the absence of cytokines. Surprisingly, both *Tet1*⁻/⁻, *Tet2*⁻/⁻ DKO and *Tet1*⁻/⁻, *Tet2*⁻/⁻, *Tet3*⁻/⁻ TKO lines yielded SSEA1⁺/CD61⁺ cells at similar efficiencies regardless of the presence or absence of cytokines, thereby outperforming *Otx2*-null cells in this regard (Fig. 5B,D). This high efficiency PGCLC differentiation was confirmed by immunofluorescence staining for the key germline TF AP2γ. The proportion of AP2γ positive cells within day 2 aggregates was significantly higher in *Tet1*⁻/⁻, *Tet2*⁻/⁻, *Tet3*⁻/⁻ TKO aggregates than in wild-type aggregates, both in the presence and absence of cytokines (Fig. 5E), agreeing with the increased germline entry suggested by the SSEA1/CD61 profiles. Importantly, we confirmed cytokine-independent PGCLC differentiation in a second TKO clone derived from E14Tg2a (TKO C19, see Fig. EV3A,B) as well as in a completely independent triple-knockout line derived by another laboratory (Ginno et al, 2020) (Fig. EV3C,D). We attempted to phenocopy the TET-TKO phenotype using a commercial TET inhibitor, Bobcat339 (Chua et al, 2019). This did not produce the same increase in PGCLC differentiation observed by genetic deletion of *Tet* genes (Fig. EV3E). However, the biological activity of Bobcat339 has been challenged recently (Weirath et al, 2022), which may explain its inability to mimic the phenotype of genetic ablation of TET enzymes.

We tested the global methylation level in PGCLCs derived from wild-type and *Tet1*⁻/⁻, *Tet2*⁻/⁻, *Tet3*⁻/⁻ TKO cells. Global methylation is not statistically different between wild-type PGCLCs and *Tet1*⁻/⁻, *Tet2*⁻/⁻, *Tet3*⁻/⁻ TKO PGCLCs derived in the presence or absence of cytokines (Fig. EV4A). Interestingly, hydroxymethylation is high in wild-type PGCLCs compared to wild-type somatic cells differentiated without cytokines and is undetectable in *Tet1*⁻/⁻, *Tet2*⁻/⁻, *Tet3*⁻/⁻ TKO PGCLCs. These results suggest that hydroxymethylation is not critical for the formation of PGCLCs (Fig. EV4A).

To further assess the PGCLC identity suggested by SSEA1/CD61 expression, we performed RT-qPCR of key germline and germ layer markers on sorted SSEA1⁺/CD61⁺ cells. At day 6 *Tet1*⁻/⁻, *Tet2*⁻/⁻, *Tet3*⁻/⁻ TKO cells do not express the germ layer markers *Cdh2* (ectoderm), *Col1a1* (mesoderm) or *Foxa2* (endoderm) (Fig. EV4B). Instead, regardless of the addition of cytokines day 6 *Tet1*⁻/⁻, *Tet2*⁻/⁻, *Tet3*⁻/⁻ TKO PGCLCs express the germline markers *Dppa3* and *Ddx4* (*Vasa*) (Fig. EV4C). Interestingly, compared to wild-type PGCLCs, *Tet1*⁻/⁻, *Tet2*⁻/⁻, *Tet3*⁻/⁻ TKO PGCLCs express lower *Dppa3* levels and higher *Ddx4* levels (Fig. EV4C). These results suggest that cells can express the germline programme without TET protein function, but that there may be some transcriptional differences between wild-type and *Tet1*⁻/⁻, *Tet2*⁻/⁻, *Tet3*⁻/⁻ PGCLCs.

Together, these results suggest that the combined loss of TET1 and TET2 proteins enables enhanced PGCLC differentiation and,

## A

**Self-renewal assay +LIF**

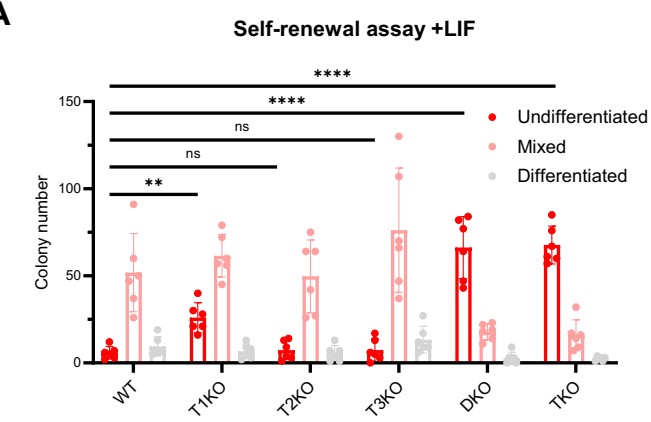

- Undifferentiated
- Mixed
- Differentiated

## B

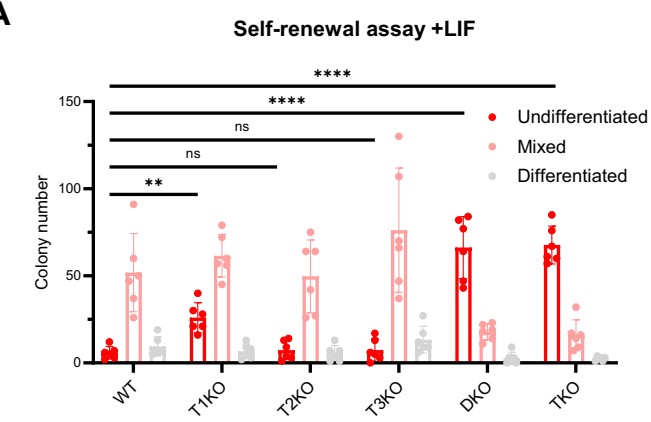

WT　　Tet1 $^{-/-}$　　Tet2 $^{-/-}$　　Tet3 $^{-/-}$　　Tet1 $^{-/-}$, Tet2 $^{-/-}$　　Tet1 $^{-/-}$, Tet2 $^{-/-}$, Tet3 $^{-/-}$

## C

WT　　Tet1 $^{-/-}$　　Tet2 $^{-/-}$　　Tet3 $^{-/-}$　　Tet1 $^{-/-}$, Tet2 $^{-/-}$　　Tet1 $^{-/-}$, Tet2 $^{-/-}$, Tet3 $^{-/-}$

OCT4

NANOG

OCT4 +
NANOG
(merge)

DAPI

**Figure 3. Characterisation of TET function during ESC self-renewal.**

(A) Alkaline phosphatase staining of the indicated ESC lines following plating at clonal density in serum/LIF medium for 7 days. Colonies were counted and categorised according to their alkaline phosphatase staining (data points: replicate wells, centre: mean, error bars: standard deviation, $n = 6$). Stars indicate statistical significance compared to wild-type for undifferentiated colonies (one-way ANOVA test). Individual $p$ values are provided in Table EV1. (B) For each of the genotypes quantified in (A) an image representative of the main class of colonies formed in serum/LIF is shown by phase contrast microscopy. Scale bars: 100 µm. (C) Co-immunofluorescence for OCT4 (red) and NANOG (green) in the indicated ESC lines cultured in serum/LIF. Scale bars: 50 µm.

importantly, permits this differentiation to proceed without the requirement for cytokine signalling.

## Transcriptional changes indicate efficient germline differentiation of TET-deficient cells

To determine whether the expression of germline markers observed in TET-deficient cells was reflected in overall transcriptional changes consistent with PGC differentiation, RNA-seq analysis was performed on wild-type and $Tet1^{-/-}$, $Tet2^{-/-}$, $Tet3^{-/-}$ TKO cells in ESCs, EpiLCs and during PGCLC differentiation. PGCLC samples were compared to a cell population known not to contain PGCLCs, obtained by differentiating wild-type cells without PGC-promoting cytokines. First, the *Tet* gene expression dynamics during PGCLC differentiation were analysed. In wild-type cells, *Tet1* and *Tet2* are upregulated in cultures undergoing germline differentiation in the presence of cytokines, but are lowly expressed in cultures in the absence of cytokines, where no PGCLCs form. In contrast, *Tet3* is downregulated in the presence of cytokines but upregulated in the absence of cytokines (Fig. EV5A).

Principal component (PC) analysis of the RNA-seq data shows that wild-type and $Tet1^{-/-}$, $Tet2^{-/-}$, $Tet3^{-/-}$ TKO EpiLCs cluster together (Fig. 6A). This global transcriptional analysis confirms the ability of $Tet1^{-/-}$, $Tet2^{-/-}$, $Tet3^{-/-}$ ESCs to transit to an EpiLC state (Figs. 4 and EV2C). From this common starting point, wild-type EpiLCs differentiated in the absence of cytokines move along both PCs (Fig. 6A). In contrast, wild-type EpiLCs differentiated for 6 days in the presence of PGC-promoting cytokines show the opposite variation along PC1 (Fig. 6A). Interestingly, after only 2 days $Tet1^{-/-}$, $Tet2^{-/-}$, $Tet3^{-/-}$ TKO PGCLCs are transcriptionally comparable to wild-type cells after 6 days of PGCLC differentiation (Fig. 6A), and this is the case with or without PGC-promoting cytokines (Fig. 6A). After 6 days, $Tet1^{-/-}$, $Tet2^{-/-}$, $Tet3^{-/-}$ TKO PGCLCs have diverged further along PC2, and this occurs in the presence or absence of cytokines. These results suggest that during PGCLC differentiation, $Tet1^{-/-}$, $Tet2^{-/-}$, $Tet3^{-/-}$ TKO cells undergo similar changes to wild-type PGCLCs and that these changes do not require PGC-promoting cytokines.

The above analysis suggests that by day 2, $Tet1^{-/-}$, $Tet2^{-/-}$, $Tet3^{-/-}$ TKO cells have a developmentally advanced transcriptome. To understand the global transcriptional changes behind the PCA, the expression and biological function of the top 400 genes contributing to PC1 and PC2 were analysed (Fig. 6A,B) and day 2 $Tet1^{-/-}$, $Tet2^{-/-}$, $Tet3^{-/-}$ TKO cells were directly compared to wild-type PGCLCs (Fig. EV5B–D).

The top 800 genes contributing to the PCA were grouped by hierarchical clustering, resulting in four distinct groups (Fig. 6B). Group 1 includes genes associated with "stem cell population maintenance" and "reproductive structure development", such as *Nanog*, *Prdm14*, *Tfap2c* and *Prdm1*. Group 1 shows similar levels of

expression throughout all PGCLCs (wild-type plus cytokines and $Tet1^{-/-}$, $Tet2^{-/-}$, $Tet3^{-/-}$ TKO plus and minus cytokines), but reduced levels of expression in wild-type cells differentiating without cytokines (Fig. 6B). A comparison of the transcriptional signatures supports an overlap between day 6 wild-type cells and $Tet1^{-/-}$, $Tet2^{-/-}$, $Tet3^{-/-}$ TKO cells at day 2 (Fig. EV5B,D). The majority of genes upregulated in wild-type PGCLCs at day 6 are also upregulated in $Tet1^{-/-}$, $Tet2^{-/-}$, $Tet3^{-/-}$ TKO PGCLCs as early as day 2, and this occurs irrespective of the presence or absence of cytokines (1080 genes, 76.5% of all WT genes) (Fig. EV5B). Commonly upregulated genes are mostly related to reproduction and meiosis and include key germline regulators *Prdm14*, *Prdm1* and *Tfap2c*, which are expressed at similar levels in wild-type PGCLCs at day 6 and in $Tet1^{-/-}$, $Tet2^{-/-}$, $Tet3^{-/-}$ TKO PGCLCs at day 2 (Fig. EV5C). Therefore, by day 2 $Tet1^{-/-}$, $Tet2^{-/-}$, $Tet3^{-/-}$ TKO PGCLCs have undergone the same upregulation of the germline programme that takes 6 days to occur in wild-type PGCLCs.

Groups 2 and 3 show similar expression patterns. These genes are highly expressed in wild-type cells after 6 days of differentiation without cytokines but remain low in wild-type cells differentiated with cytokines and in all $Tet1^{-/-}$, $Tet2^{-/-}$, $Tet3^{-/-}$ cells analysed, irrespective of the presence or absence of cytokines (Fig. 6B). Genes in groups 2 and 3 are associated with neural differentiation and include *Ascl1*, *Pax7*, *Neurog1* and *Neurog2* (Fig. 6B). A comparative analysis of downregulated genes showed that most genes downregulated in wild-type PGCLC were repressed in $Tet1^{-/-}$, $Tet2^{-/-}$, $Tet3^{-/-}$ PGCLCs by day 2, and again that this occurred irrespective of the presence or absence of cytokines (Fig. EV5D). Commonly downregulated genes include neural markers like *Neurog2 and Neurod4* (Fig. EV5C), and associated gene ontology terms are mostly related to the neural fate (Fig. EV5D). This suggests that in the absence of cytokines wild-type cells follow a neural fate, which is repressed in $Tet1^{-/-}$, $Tet2^{-/-}$, $Tet3^{-/-}$ TKO cells regardless of the addition of cytokines.

Overall, these results suggest that by day 2, $Tet1^{-/-}$, $Tet2^{-/-}$ and $Tet3^{-/-}$ TKO cells differentiated in the presence or absence of cytokines both upregulate the germline programme and repress the somatic programme.

## TET-deficient PGCLCs show precocious expression of late germline markers

A fourth group of genes are expressed highly in $Tet1^{-/-}$, $Tet2^{-/-}$, $Tet3^{-/-}$ TKO PGCLCs after 6 days of differentiation (with or without cytokines) (Fig. 6B). This group differs from group 1 by being expressed at lower levels in wild-type PGCLCs at day 6 than in $Tet1^{-/-}$, $Tet2^{-/-}$, $Tet3^{-/-}$ TKO PGCLCs. This reflects PC differences between wild-type and $Tet1^{-/-}$, $Tet2^{-/-}$, $Tet3^{-/-}$ TKO cells at day 6 (Fig. 6A,B). Gene ontology (GO) enrichment shows

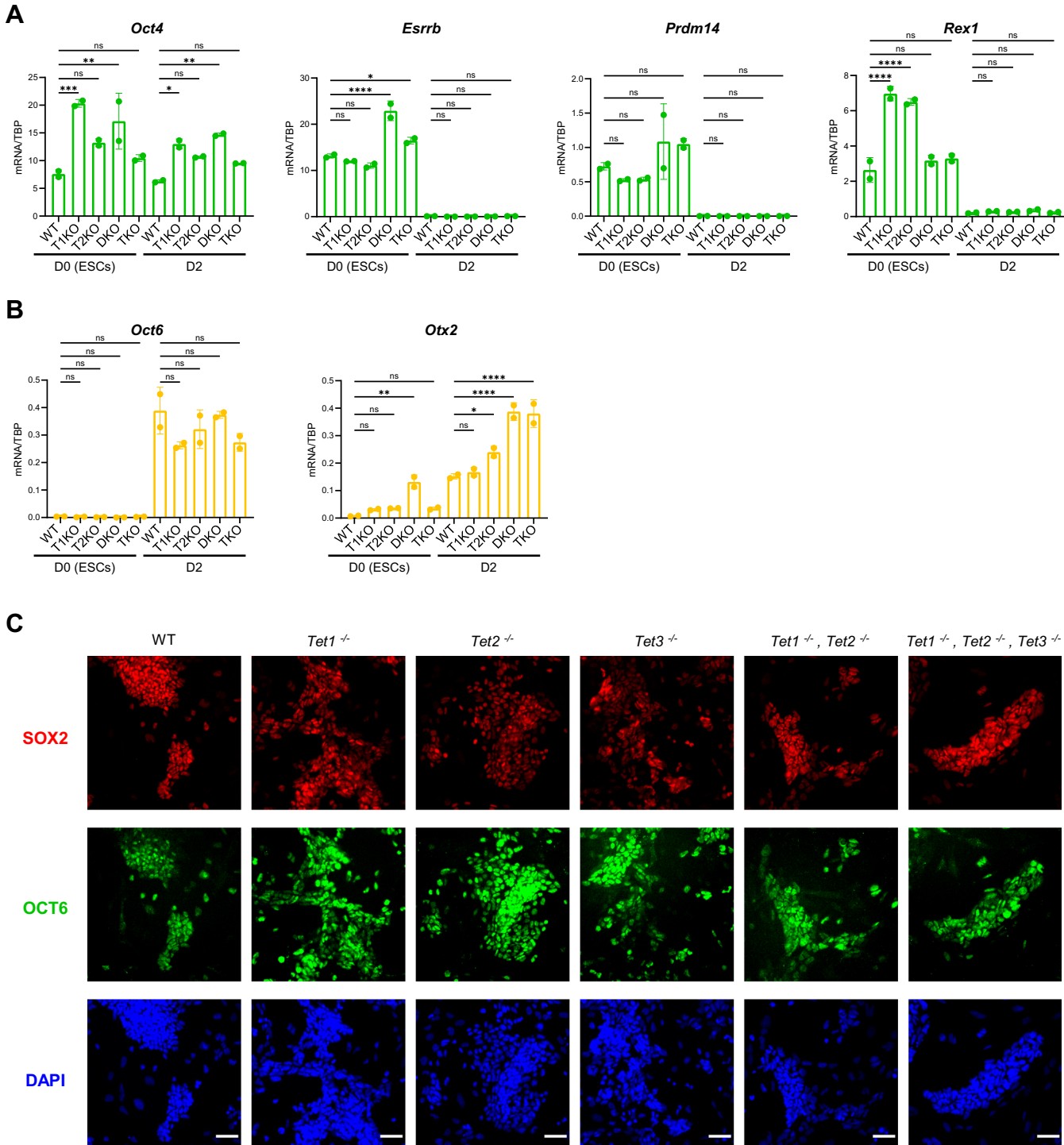

**Figure 4. Characterisation of TET function during the transition from naïve to formative and primed pluripotency.**

(A, B). mRNA levels of naïve (A) and primed (B) pluripotency factors in the indicated cell lines cultured as ESCs in 2i/LIF (D0) or after EpiLC differentiation (D2). mRNA levels were quantified by RT-qPCR and normalised to TBP mRNA levels (data points: technical replicates, centre: mean, error bars: standard deviation, $n = 2$). Stars indicate statistical significance compared to wild-type (one-way ANOVA test). Individual $p$ values are provided in Table EV1. (C) Co-immunofluorescence for SOX2 (red) and OCT6 (green) in the indicated EpiSC lines (passage 11) cultured in activin/FGF. Scale bars: 50 µm.

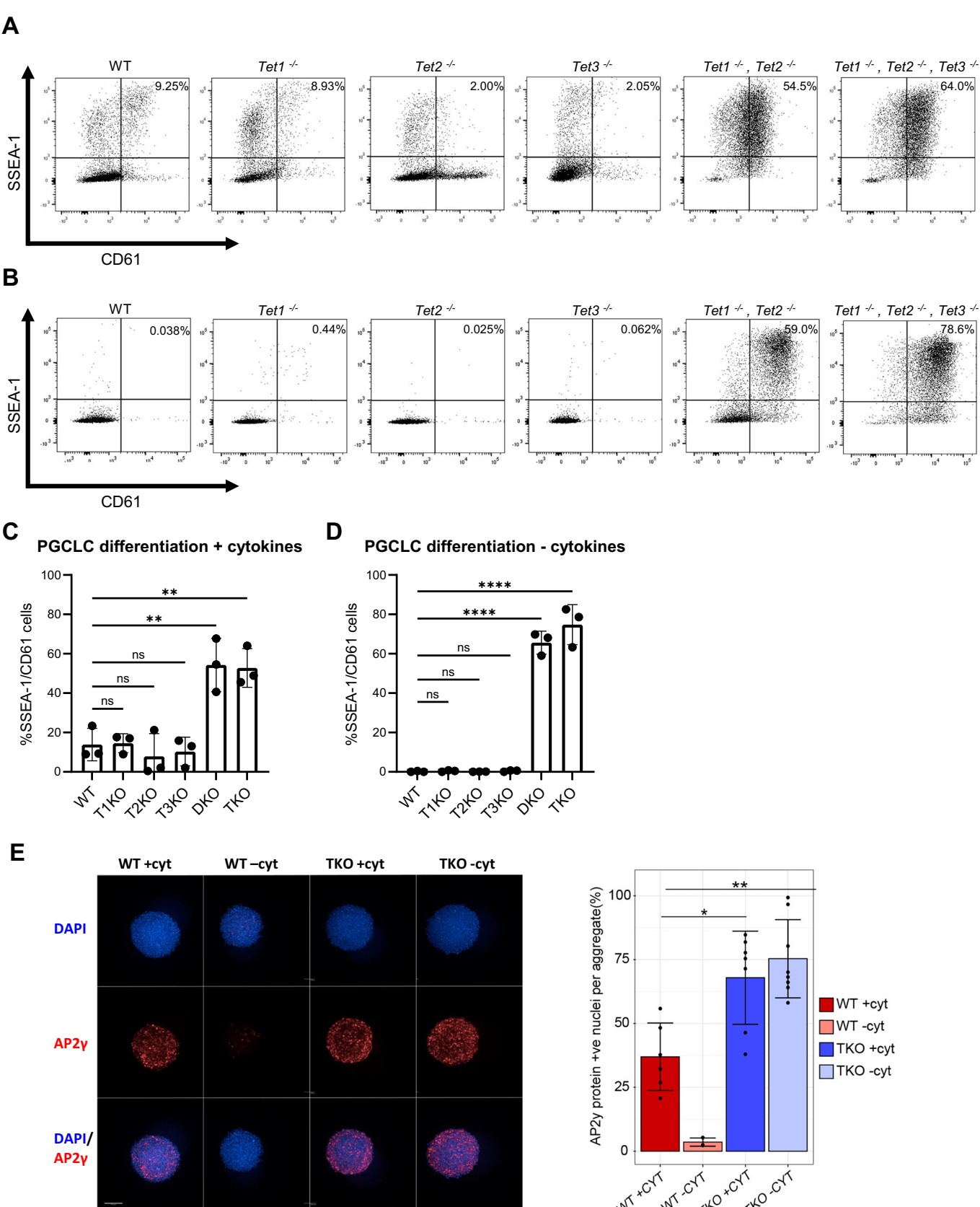

Figure 5. Characterisation of TET function during germline commitment.

(A, B) Flow cytometric analysis of the indicated cell lines following 6 days of PGCLC differentiation with (A) or without (B) PGC-promoting cytokines. SSEA-1/CD61 double-positive cells in the top right quadrant report PGCLCs. (C, D) Quantification of SSEA-1/CD61 double-positive cells following 6 days of PGCLC differentiation with (C) or without (D) cytokines in the indicated cell lines (data points: replicate experiments, centre: mean, error bars: standard deviation, $n = 3$). Stars indicate statistical significance compared to wild-type (one-way ANOVA test). Individual $p$ values are provided in Table EV1. (E) Immunofluorescence of AP2γ protein in day 2 PGCLC aggregates. Representative images (left) are the maximal projection of 3 Z planes of the same aggregate. Quantification (right) of percentage of nuclei positive for AP2γ in each aggregate (data points: individual aggregates, centre: mean, error bars: standard deviation, biological replicates = 2). Stars indicate statistical significance compared to wild-type + cytokines (Student's $t$-test, * ≤0.05, ** ≤0.01). Individual $p$ values are provided in Table EV1.

that genes in group 4 are mostly associated with the gene ontology terms "DNA methylation" which includes *Mov10l1, Dppa3, Mael* and *Wt1*, genes previously associated with germline or gonad development (Sato et al, 2002; Wilhelm and Englert, 2002; Soper et al, 2008; Hackett et al, 2012; Vourekas et al, 2015) and "cellular process involved in reproduction" which includes the late germline markers *Stra8, Dazl* and *Taf7l* (Fig. 6B). Indeed, temporal assessment shows an increased expression of *Stra8, Dazl, Taf7l* and *Spesp1* mRNAs in *Tet1⁻/⁻, Tet2⁻/⁻, Tet3⁻/⁻* TKO PGCLCs as early as day 2 (Fig. EV5E).

### Expression of PGC TFs indicates accelerated germline entry in TET-deficient cells

The expression of late markers suggests that the transcriptome of *Tet1⁻/⁻, Tet2⁻/⁻, Tet3⁻/⁻* TKO PGCLCs might reflect a later stage in PGC development, consistent with the accelerated germ cell differentiation pinpointed by the PC analysis. To assess whether the similarity between day 2 *Tet1⁻/⁻, Tet2⁻/⁻, Tet3⁻/⁻* TKO PGCLCs and day 6 wild-type PGCLCs could be explained by accelerated germline entry, the expression of *Tfap2c, Prdm1* and *Prdm14* was examined at 12-hourly intervals during the first 3 days of PGCLC differentiation. Following the addition of cytokines, all three mRNAs show increased expression in *Tet1⁻/⁻, Tet2⁻/⁻, Tet3⁻/⁻* TKO cells over the first 2 days of differentiation, compared to a more delayed upregulation in wild-type cells (Fig. 6C). In particular, *Tfap2c* and *Prdm14* mRNAs are more highly expressed in *Tet1⁻/⁻, Tet2⁻/⁻, Tet3⁻/⁻* TKO cells between 24 and 48 h (Fig. 6C).

### TET-deficient PGCLCs express a transcriptional profile characteristic of a later developmental stage in vivo

To assess how these specific changes relate to transcriptomic changes during development, samples were compared to in vivo germ cell data at E9.5, E12.5 and E14.5, and in vitro EpiLCs generated using single-cell 3 prime RNA-seq (SC3-seq) by (Ishikura et al, 2021; Miyauchi et al, 2017). PC analysis reveals that technical differences account for variability along PC1, with samples clustering on opposite sides of PC1 based on the technique used (Fig. EV5F). However, PC2 reflects interesting biological differences. In vitro samples generated in this study align across time, with EpiLCs at the top, day 6 wild-type PGCLCs and day 2 *Tet1⁻/⁻, Tet2⁻/⁻, Tet3⁻/⁻* TKO PGCLCS in the middle, and day 6 *Tet1⁻/⁻, Tet2⁻/⁻, Tet3⁻/⁻* TKO PGCLCs at the bottom (Fig. EV5F). On the right side, data from (Ishikura et al, 2021) follows a similar pattern, with EpiLCs aligning with our EpiLCs in PC2, and germ cells following a similar temporal pattern from E9.5 to E14.5. Interestingly, day 6 wild-type PGCLCs and day 2 *Tet1⁻/⁻, Tet2⁻/⁻,*

*Tet3⁻/⁻* TKO PGCLCs align with E12.5 germ cells within PC2, while day 6 *Tet1⁻/⁻, Tet2⁻/⁻, Tet3⁻/⁻* TKO PGCLCs align with E14.5 germ cells (Fig. EV5F). This suggests that the transcriptome of day 6 *Tet1⁻/⁻, Tet2⁻/⁻, Tet3⁻/⁻* TKO PGCLCs corresponds to a later developmental stage.

### TET-deficient cells contribute to the germline in vivo

To determine how TET-deficient cells perform in vivo, single-cell data from chimeric embryos made by injecting either wild-type or TET-TKO mutant cells into 4n embryos were compared (Cheng et al, 2022). By reanalysing this data, we aimed to determine whether TET-TKO cells can contribute to the germline in vivo. Cell identities determined by mapping to a gastrulation atlas were used to calculate the relative composition of cell types per embryo (Cheng et al, 2022). The relative content of PGCs appears, on average, slightly higher in TET-TKO mutants than in wild-type chimeras, although this difference is not significant (Fig. EV6A). PGCs were then identified through clustering and marker expression. This revealed two clusters expressing germline markers. A nascent PGC cluster expresses early PGC markers, such as *Dppa3, Prdm1* and *Prdm14*, while a late PGC cluster expresses later PGC transcripts (*Nanos3, Dazl* and *Stra8*) (Fig. EV6B). Importantly, while no significant difference is detected in the nascent PGC population, the proportion of late PGCs per embryo is significantly higher in TET-TKO compared to wild-type chimeras (Fig. EV6C). This shows that while TET-TKO cells fail to contribute to a number of somatic tissues in chimeras (Cheng et al, 2022), they do contribute to the PGC population, and with a higher efficiency than wild-type cells, thereby complementing our in vitro findings.

Taken together, this data shows that germline markers are expressed at similar levels in *Tet1⁻/⁻, Tet2⁻/⁻, Tet3⁻/⁻* TKO PGCLCs and wild-type PGCLCs, but that somatic markers upregulated in wild-type cells in the absence of cytokines remain low in *Tet1⁻/⁻, Tet2⁻/⁻, Tet3⁻/⁻* TKO PGCLCs throughout differentiation. Moreover, *Tet1⁻/⁻, Tet2⁻/⁻, Tet3⁻/⁻* TKO PGCLCs enter the germline at an accelerated rate. The major differences between *Tet1⁻/⁻, Tet2⁻/⁻, Tet3⁻/⁻* TKO PGCLCs and wild-type PGCLCs at day 6 are in mRNAs characteristic of later germline development; differences that are not evident in day 2 *Tet1⁻/⁻, Tet2⁻/⁻, Tet3⁻/⁻* TKO PGCLCs. Alongside the earlier upregulation of *Prdm14, Tfap2c* and *Prdm1* mRNAs, this suggests precocious germline development of *Tet1⁻/⁻, Tet2⁻/⁻, Tet3⁻/⁻* TKO cells.

## Discussion

Here, we have compared ESC lines lacking one or several *Tet* alleles. This allowed us to characterise unique and redundant functions of TET proteins and to show the early cellular changes

**A**

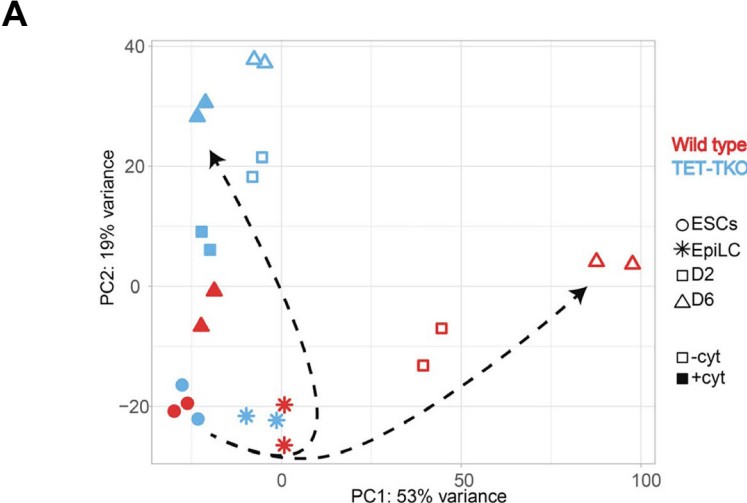

**B**

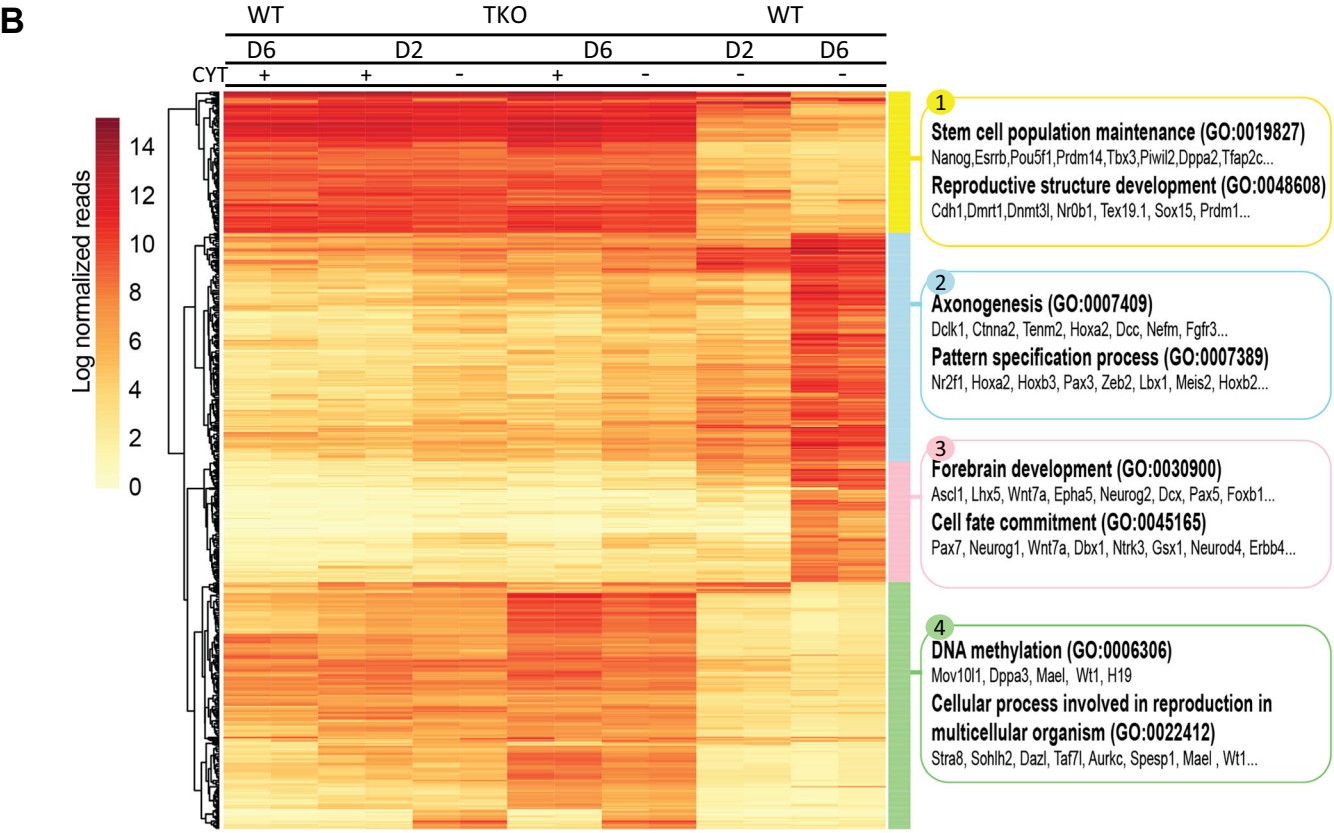

**C**

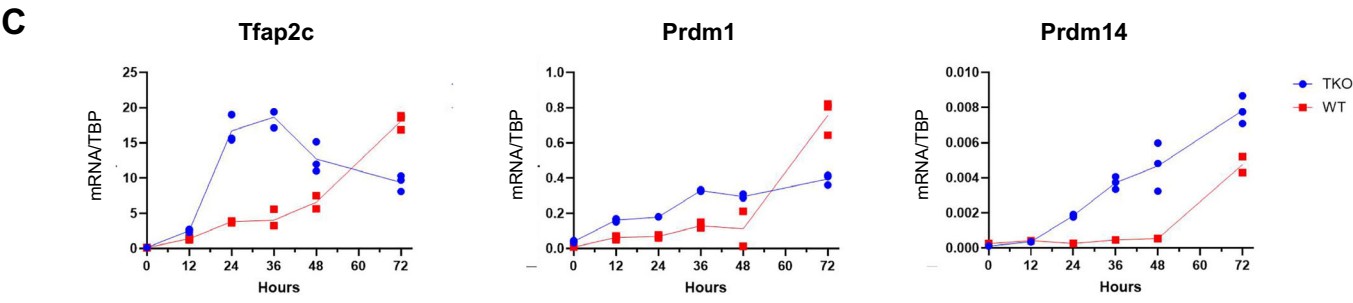

**Figure 6.   TET-TKOs upregulate germline markers in an accelerated manner.**

(A) Principal component (PC) analysis of bulk RNA-seq samples. Wild-type and TKO cells at ESCs, EpiLCs, day 2 and day 6 of differentiation (+/− cytokines). Wild-type cells differentiated in the presence of cytokines were sorted into SSEA1+/CD61+ populations to select for PGCLCs. All other populations were collected as full aggregates (n = 2). (B) Heatmap of the log-normalised reads of the top variable genes across all samples (top 400 genes contributing to PC1 and top 400 genes contributing to PC2). Genes are grouped by hierarchical clustering and GO terms are shown for each group (right-hand side). (C) Expression of key germline markers measured at the indicated times of PGCLC differentiation in the presence of cytokines. mRNA levels were quantified by RT-qPCR and normalised to TBP mRNA levels. Data are from a representative of three independent experiments (data points: technical replicates, line: mean, error bars: standard deviation, n = 3).

during distinct differentiation contexts. This uncovered key redundant roles for TET1 and TET2 in supporting somatic lineage differentiation and acting as a critical barrier to germline induction.

In agreement with previous literature (Dawlaty et al, 2011, 2013, 2014; Lu et al, 2014; Hu et al, 2014; Verma et al, 2018), all *Tet* knockout ESC lines have a normal morphology and express similar levels of pluripotency markers to wild-type ESCs during routine passaging in serum/LIF. Furthermore, TET-deficient ESCs are defective in differentiation into somatic lineages and maintain expression of pluripotency genes. Both a block to differentiation and retention of pluripotency were previously observed in triple *Tet* knockout cells and were attributed to redundant activities of TET1, TET2 and TET3 (Dawlaty et al, 2014; Verma et al, 2018). However, we show here that *Tet1*−/−, *Tet2*−/− DKO and *Tet1*−/−, *Tet2*−/−, *Tet3*−/− TKO ESCs have similar blocks to differentiation. Therefore, redundancy amongst TET proteins in driving loss of pluripotency is restricted to TET1 and TET2 and does not extend to TET3.

*Tet1*−/−, *Tet2*−/− DKO and *Tet1*−/−, *Tet2*−/−, *Tet3*−/− TKO ESCs have more undifferentiated colony morphologies than wild-type or single *Tet* knockout cells when plated at clonal density in LIF. This appears to result from an enhanced sensitivity to LIF, rather than LIF independence, since undifferentiated colonies do not form when mutant cell lines are plated at clonal density in the absence of exogenous LIF. In addition, *Tet1*−/−, *Tet2*−/− DKO and *Tet1*−/−, *Tet2*−/−, *Tet3*−/−· TKO lines can form "ESC-like" colonies after prolonged culture at high density in the absence of exogenously added LIF in either serum-free monolayer differentiation media, or during aggregation-induced differentiation. These results may be due to an enhanced responsiveness of *Tet1*−/−, *Tet2*−/− DKO and *Tet1*−/−, *Tet2*−/−, *Tet3*−/− TKO cells to endogenously produced LIF accumulating in such high-density cultures (Chambers et al, 2003). This enhanced LIF sensitivity could be connected to the recent observation that TET1 and TET2 are required for pluripotent cells to survive embryonic diapause (Stötzel et al, 2024), a process that also requires signalling through the LIFR/gp130 pathway (Nichols et al, 2001). Interestingly, studies in hematopoietic cells also suggest that TET1 and TET2 may act downstream of STAT signalling (Jiang et al, 2017; Xue et al, 2024). Therefore, investigating the exact nature of the relationship between TET proteins and LIF signalling in pluripotent cells will be an interesting area for future investigations.

Our analysis also extends prior studies by characterising redundant TET function during pluripotent state transitions and during commitment to the germline. We demonstrated that TET-deficient ESCs can differentiate normally into EpiLCs or EpiSCs, which capture in vitro the transitions between naïve, formative and primed pluripotent states that occur between pre-

and post-implantation epiblast development (Smith, 2017; Posfai et al, 2021). These results are consistent with a previous study (Parry et al, 2023) and offer an explanation for the in vivo phenotype of *Tet1*−/−, *Tet2*−/−, *Tet3*−/− TKO mouse embryos which are morphologically indistinguishable from wild-type until the onset of gastrulation at ~E6.5 (Dai et al, 2016; Li et al, 2016).

TET proteins have not previously been implicated in the initial induction of PGCs from the post-implantation epiblast. While loss of TET1 is known to impair demethylation at imprinted genes and meiotic genes, this does not occur until ~E13.5, many days after germline specification (Yamaguchi et al, 2012, 2013; Hackett et al, 2013; Hill et al, 2018). It was therefore unexpected that *Tet1*−/−, *Tet2*−/− DKO and *Tet1*−/−, *Tet2*−/−, *Tet3*−/− TKO cell lines should have an increased propensity to form PGCLCs during in vitro differentiation. Of note, TET proteins appear to be expressed in the entire epiblast of post-implantation embryos around E6.5-E7.5 (Dai et al 2016; Khoueiry et al, 2017) and therefore do not discriminate between soma and germline. The increased PGCLC differentiation of TET-deficient cells might indicate a role for TET proteins to actively prevent germline commitment. However, it is difficult to rationalise such a role with the increasing expression of *Tet1* and *Tet2* mRNAs during the initial days of differentiation of EpiLCs to PGCLCs. Alternatively, this phenotype could be an indirect consequence of a failure of *Tet1*−/−, *Tet2*−/− DKO and *Tet1*−/−, *Tet2*−/−, *Tet3*−/− TKO cells to differentiate into somatic lineages. In this case, TET proteins could promote activation of somatic genes by demethylation of enhancers (Charlton et al, 2020; Ginno et al, 2020).

Remarkably, the enhanced germline differentiation of TET-deficient cells occurs in the absence of the otherwise requisite PGC-promoting cytokines. PGC-promoting cytokines act by rapidly downregulating *Otx2* mRNA soon after initiation of PGCLC differentiation from EpiLCs (Zhang et al, 2018a; Zhang and Chambers, 2019). Consequently, *Otx2*-null cells also differentiate to PGCLCs without cytokines (Zhang et al, 2018a). EpiLCs can also differentiate to PGCLCs without cytokines upon induced expression of NANOG (Murakami et al, 2016; Zhang et al, 2018b). As NANOG can also repress *Otx2* mRNA expression (Vojtek et al, 2022), the TET-TKO PGCLC phenotype could occur if *Otx2* mRNA was not upregulated during the ESC-EpiLC transition. However, our analysis shows that TET-deficient EpiLCs have upregulated *Otx2* mRNA. Alternatively, the similarity between the phenotypes of the *Tet1*−/−, *Tet2*−/− DKO, *Tet1*−/−, *Tet2*−/−, *Tet3*−/− TKO and *Otx2*−/− lines may result from the OTX2-mediated targeting of TET proteins to somatic regulatory elements that subsequently become demethylated and active. In this scenario, ectopic OTX2 expression would not be able to rescue the TET

mutant somatic block. This will be an interesting experiment for the future.

Since the lack of DNA demethylating enzymes (TETs) reported here causes a block in somatic differentiation associated with a dramatic increase in PGCLC differentiation, one might expect a lack of methylating enzymes (DNMTs) to have the opposite effect and to fully block PGCLC differentiation. However, this simple relationship does not hold. Complete loss of DNA methylation ($Dnmt1^{-/-}$, $Dnmt3a^{-/-}$, $Dnmt3b^{-/-}$) in ESCs also causes severe defects in somatic differentiation (Tsumura et al, 2006; Sakaue et al, 2010; Schmidt et al, 2012), but has distinct molecular phenotypes compared to TET-deficient cells (Parry et al, 2023). In addition, a recent study showed that loss of DNA methylation does not increase the number of differentiating PGCLCs but extends the period of germline competence beyond 48 h of EpiLC differentiation (Schulz et al, 2024). DNMT TKO EpiLCs can also differentiate into PGCLCs in the absence of cytokines, but at an efficiency of only ~5%, far less than observed here for $Tet1^{-/-}$, $Tet2^{-/-}$, $Tet3^{-/-}$ TKO cells. Together, these results support the idea that DNA methylation (and methylation modifications) play cryptic roles in the induction of correct cell fates in somatic and germline differentiation. Interestingly, a recent report has proposed that both DNMTs and TETs may be required for cyclic DNA methylation and demethylation (Parry et al, 2021) as a way of providing epigenetic marks for epigenetic "readers" that may influence cell fate decisions (Spruijt et al, 2013; Iurlaro et al, 2013; Song et al, 2021). Alternatively, such 'futile cycles' can allow rapid alterations in regulatory circuits during cell fate decisions (Samoilov et al, 2005). However, the differences in phenotype between cells that are unable to methylate DNA and cells that cannot oxidise methylated DNA also suggest quite distinct regulatory roles for intermediates in the methylation cycle in dictating choice between soma and germline. Moreover, whether any of these considerations relate to the precocious germline development that we report here for $Tet1^{-/-}$, $Tet2^{-/-}$, $Tet3^{-/-}$ TKO cells will be an important question for future studies. Alternatively, biological functions of TET proteins that are independent of their enzymatic activity and that occur via recruitment of the co-repressor Sin3 (Zhu et al, 2018; Chrysanthou et al, 2022; Stolz et al, 2022; Flores et al, 2023) or interplay with the Polycomb machinery (van der Veer et al, 2023; Chrysanthou et al, 2022; Huang et al, 2022) may be important in determining cell fate choice. Deciphering the exact mechanism by which TET proteins control gene expression at the juncture between the soma and the germline will be an important area for future investigation.

# Methods

### Reagents and tools table

| Reagent/resource | Reference or source | Identifier or catalogue number |
| --- | --- | --- |
| **Experimental models** | | |
| E14Tg2a (*M. musculus*) | Hooper et al, 1987 | N/A |
| TET1 KO (*M. musculus*) | This study | N/A |
| TET2 KO (*M. musculus*) | This study | N/A |
| TET3 KO (*M. musculus*) | This study | N/A |
| TET1/2 DKO (*M. musculus*) | This study | N/A |
| TET1/2/3 TKO (*M. musculus*) | This study | N/A |
| Dnmt3a/b Floxed ESCs (*M. musculus*) | Ginno et al, 2020 | N/A |
| TET1/2/3 TKO (*M. musculus*) | Ginno et al, 2020 | N/A |
| **Recombinant DNA** | | |
| Cas9-T2A-EGFP and sgRNA co-expression plasmid (empty backbone) | Addgene | #48138 |
| Cas9-T2A-mCherry and sgRNA co-expression plasmid (empty backbone) | This study | N/A |
| **Antibodies** | | |
| Goat anti-OCT4 | Santa Cruz | sc-8628 |
| Rat anti-NANOG | eBioscience | 14-5761-80 |
| Rabbit anti-SOX2 | Abcam | ab92494 |
| Goat anti-OCT6 | Santa Cruz | sc-11661 |
| Mouse anti-AP2γ | Santa Cruz | sc12762 |
| Mouse anti-TUJ1 | Biolegend | 801202 |
| Alexa Fluor 647 Mouse anti-CD15 (SSEA-1) | Biolegend | 125608 |
| Phycoerythrin (PE) Hamster anti-CD61 | Biolegend | 104307 |
| **Oligonucleotides and other sequence-based reagents** | | |
| gRNAs for CRISPR/Cas9 engineering | This study | See the list of gRNAs in the Methods section |
| Genotyping primers | This study | See the list of genotyping primers in the Methods section |
| RT-qPCR primers | This study | See the list of RT-qPCR primers in the Methods section |
| **Chemicals, enzymes and other reagents** | | |
| Glasgow minimum essential medium | Sigma | G5154 |
| L-glutamine | Invitrogen | 25030-024 |
| 1x pyruvate solution | Invitrogen | 11360-039 |
| MEM non-essential amino acids | Invitrogen | 11140-035 |
| 2-Mercaptoethanol | Gibco | 31350010 |
| Accutase | StemPro | A1110501 |
| DMEM:F12 | GIBCO | 12634010 |
| Neurobasal | GIBCO | 21103049 |
| poly-L-ornithine | Sigma | P3655 |
| Laminin | BD Biosciences | 354232 |
| Fibronectin | Merck | FC010 |
| Knock-out Serum Replacement (KOSR) | Gibco | 10828-028 |

| Reagent/resource | Reference or source | Identifier or catalogue number |
|---|---|---|
| Lipofectamine 3000 | Thermo Fisher Scientific | L3000008 |
| DNA extraction kit | Qiagen | 69506 |
| Proteinase K | Invitrogen | 100005393 |
| 4% PFA | Thermo fisher | J19943 |
| Triton X-100 | Sigma | X100 |
| Low-melting point UltraPure agarose | Thermo Fisher | 16500500 |
| RapiClear 1.47 | Sunjin lab | RC147001 |
| RNeasy Plus Mini kit | Qiagen | 74136 |
| SuperScript III | Invitrogen | 18080044 |
| Takyon SYBR MasterMix | Eurogentec | UF-NSMT-B0701 |
| Direct-zol RNA MiniPrep kit | Zymo Research | R2052 |
| ESGRO LIF | Merck | ESG1106 |
| Human Fgf basic | R&D Systems | 233-FB-025/CF |
| Human Activin A | PeproTech | 120-14E |
| Recombinant human BMP-4 | Qkine | Qk038 |
| Recombinant Human BMP-8a | R&D Systems | 1073-BP-010 |
| Recombinant Mouse SCF Protein | R&D Systems | 455-MC-010 |
| Recombinant Mouse EGF Protein, CF | R&D Systems | 2028-EG-200 |
| Bobcat339 | Merck | SML2611 |
| Agencourt RNAClean XP beads | Beckman Coulter | 66514 |
| NEBNext rRNA Depletion Kit (Human/Mouse/Rat) | New England Biolabs | E6310 |
| NEBNext Ultra II Directional RNA Library Prep Kit | New England Biolabs | E7760 |
| **Software** | | |
| GraphPad Prism | https://www.graphpad.com/ | |
| Fiji: ImageJ | https://imagej.net/software/fiji/ | |
| Harmony High-Content Imaging and Analysis Software | https://www.revvity.com/gb-en/product/harmony-5-2-office-revvity-hh17000019 | |
| FlowJo | https://www.flowjo.com/ | |
| STAR 2.7.1a | (Dobin et al, 2013) | |
| Deseq2 | (Love et al, 2014) | |
| clusterProfiler | (Yu, 2024) | |
| Seurat_5.0.0 | (Hao et al, 2024) | |
| SingleCellExperiment_1.30.1 | (Amezquita et al, 2020) | |
| scater_1.36.0 | (McCarthy et al, 2017) | |
| harmony_1.2.3 | (Korsunsky et al, 2018) | |

| Reagent/resource | Reference or source | Identifier or catalogue number |
|---|---|---|
| **Other** | | |
| Roche LightCycler 480 instrument (qPCR) | Roche | |
| BD LSR Fortessa instrument (Flow cytometry) | BD | |
| Nikon Ti-E microscope | Nikon | |
| Opera Phenix microscope | Revvity | |
| Illumina NextSeq 2000 sequencer | Illumina | |

## Cell culture

All the cell lines generated in this study were derived from E14Tg2a (Hooper et al, 1987). We also obtained an independent TET triple knockout cell line (and its parental cell line: Dnmt3a/b Floxed ESCs) from Dirk Schübeler's laboratory (FMI, Basel) (Ginno et al, 2020). All cell lines were incubated in a 37 °C, 7% $CO_2$ incubator. Cells were routinely tested for mycoplasma contamination. ESCs were routinely cultured in serum/LIF medium on gelatin-coated plates. Composition of the serum/LIF medium: Glasgow minimum essential medium (GMEM; Sigma, cat. G5154), 10% foetal bovine serum, 1× L-glutamine (Invitrogen, cat. 25030-024), 1x pyruvate solution (Invitrogen, cat. 11360-039), 1× MEM non-essential amino acids (Invitrogen, cat. 11140-035), 0.1 mM 2-Mercaptoethanol (Gibco ref. 31350010), 100 U/ml LIF (made in-house).

Monolayer neuronal differentiation was performed as described in (Pollard et al, 2006; Ying et al, 2003). ESCs, previously grown in serum/LIF, were dissociated using Accutase (StemPro, cat. A1110501), washed and resuspended in N2B27 medium. 100,000 cells were plated into a well of a six-well plate coated with gelatin and containing N2B27 medium. The medium was changed every 2 days until analysis. Composition of the N2B27 medium: 50 ml DMEM:F12 (GIBCO cat. 12634010), 50 ml Neurobasal (GIBCO cat. 21103049), 1 ml 100x glutamine (Invitrogen cat. 25030-024), 1 ml 100x MEM non-essential amino acids (Invitrogen, cat. 11140-036), 100 μl 0.1 M 2-mercaptoethanol, 1 ml 100x N2 supplement (GIBCO cat. 17502048), 2 ml 50x B27 supplement (GIBCO cat. 17504044).

Embryoid bodies (EBs) were spontaneously formed by plating $5 \times 10^6$ ESCs in non-adherent 5 cm dishes with GMEM medium containing 10% foetal bovine serum for 6 days. EBs were differentiated by plating on gelatin-coated wells of a 24-well plate (2 to 8 EBs per well) in GMEM medium without LIF. Differentiated EBs were checked by brightfield imaging every day. In particular, the appearance of spontaneously beating clusters of cells (indicating cardiomyocyte differentiation) was noted for each well.

EpiSC lines were derived by in vitro differentiation from ESCs (Guo et al, 2009). About $3 \times 10^4$ ESCs were plated in a well of a six-well plate with serum/LIF medium (see composition above). After 24 h, the medium was switched to N2B27 medium (see composition above) supplemented with 20 ng/ml Human Activin A (PeproTech, cat. 120-14E) and 10 ng/ml Human Fgf basic

(R&D Systems, cat. 233-FB-025/CF). Cells were submitted to daily media changes and passaged at day 5 of the protocol in six-well plates coated with 7.5 μg/mL bovine fibronectin. Cells were maintained in N2B27 medium supplemented with Activin/Fgf and passaged every 2–3 days. Homogeneous EpiSCs were derived within ten passages.

For 2i/LIF ESC culture, N2B27 medium was supplemented with 0.4 μM PD0325901 (Axon, cat. 1408), 3 μM CHIR99021 (Axon, cat. 1386) and 1000 U/ml ESGRO LIF (Millipore, cat. ESG1106). EpiLC differentiation and PGCLC induction were performed as previously described (Hayashi et al, 2011; Hayashi and Saitou, 2013; Zhang et al, 2018a). ESCs were adapted to 2i/LIF culture for at least three passages on poly-L-ornithine (Sigma, cat. P3655) and laminin (BD Biosciences, cat. 354232) coated wells of a six-well plate. To obtain EpiLC, $1.0 \times 10^5$ ESCs were plated on a well of a 12-well plate pretreated with 16.6 μl/ml fibronectin (Millipore, cat. FC010) and containing EpiLC medium: N2B27 medium (see composition above) supplemented with 20 ng/ml Human Activin A (PeproTech, cat. 120-14E), 12 ng/ml Human Fgf basic (R&D Systems, cat. 233-FB-025/CF) and 1% knock-out serum replacement (KOSR; Gibco, cat. 10828-028). After 2 days, EpiLCs were collected for analysis or submitted to PGCLC differentiation.

To obtain PGCLCs, $1.5 \times 10^5$ EpiLCs were resuspended in 5 ml $(3 \times 10^4$ cells/ml) of GK15 medium (GMEM supplemented with 15% KOSR) supplemented with 50 ng/ml bone morphogenetic protein (BMP) 4 (Qkine, cat. Qk038), 50 ng/ml BMP-8a (R&D Systems, cat. 1073-BP-010), 10 ng/ml stem cell factor (SCF, R&D System, cat. 455-MC-010), 10 ng/ml epidermal growth factor (EGF, R&D Systems, cat. 2028-EG-200), and 1000 U/ml LIF added before replating 100 μl per well of a U-bottom 96-well plate (Thermo Fisher Scientific, cat. 268200). For cytokine-free differentiation, EpiLCs were similarly dissociated and resuspended in 5 ml of GK15 medium $(3 \times 10^4$ cells/ml) without the aforementioned cytokines. Both cells with and without cytokines were cultured for 6 days before flow cytometry analysis.

For TET inhibition, E14Tg2a EpiLCs were differentiated towards PGCLCs in a medium containing 100 μM of Bobcat339 (Merck, cat. SML2611), with or without cytokines, and analysed at day 6 by flow cytometry.

## Flow cytometry

Flow cytometry analysis was performed as previously described (Zhang et al, 2018a, 2018b). Cells were dissociated using trypsin and neutralised in PBS/10% serum. Cell pellets were collected by centrifugation and resuspended in 100 μl PBS/2% serum supplemented with Alexa Fluor 647 anti-CD15 (SSEA-1) (Biolegend, cat. 125608) and Phycoerythrin (PE) anti-CD61 (Biolegend, cat. 104307) antibodies diluted 1/200 and 1/600, respectively. Following a 20 min incubation at 4 °C, cells were washed twice in 1 ml PBS/10% serum before flow cytometry analysis on a BD LSR Fortessa instrument. Cells were first gated based on morphology (FSC/SSC), followed by selection for singlets based on linear correlations between FSC-area and FSC-height. Live cells were then gated based on exclusion of DAPI to indicate cell membrane integrity. Gates for SSEA-1 and CD61 were determined using unstained cells and cells stained with a single antibody.

## Self-renewal assays

ESCs, previously grown in serum/LIF, were collected by trypsinisation, resuspended in GMEM medium and counted. 600 cells were plated in gelatin-coated wells of a six-well plate containing complete GMEM medium with 10% foetal bovine serum (see composition above) with or without LIF. Following 7 days of culture, cells were washed in PBS and incubated for 1 min in fixative solution made by mixing 25 ml of citrate solution (18 mM citric acid, 9 mM sodium citrate, 12 mM NaCl), 8 ml of formaldehyde solution (37% v/v in water) and 65 ml of acetone. Fixed cells were washed in distilled water and stained for alkaline phosphatase (AP) expression using a leucocyte alkaline phosphatase kit (Sigma, cat. 86R-1KT). Colonies were counted and categorised according to their morphology and alkaline phosphatase staining.

## Knockout of Tet1/2/3 by CRISPR/Cas9

To knockout endogenous *Tet1/2/3* alleles, two double-strand breaks were simultaneously generated using Cas9 and two synthetic guide RNAs targeting the start and stop codon of the open reading frame of the *Tet* family gene of interest, respectively. Importantly, we checked that the excised regions contained only *Tet* family genes and no other open reading frames or functional elements annotated in the genome databases (Ensembl, UCSC Genome Browser). Guide RNAs were designed (https://www.zlab.bio/resources) and cloned into Cas9/gRNA co-expression plasmids carrying a fluorescent reporter (Addgene PX458 containing EGFP, or a custom-made version containing mCherry).

About $1 \times 10^6$ ESCs were co-transfected with two Cas9/gRNA plasmids targeting the start and stop codon of the *Tet* open reading frame, respectively, using Lipofectamine 3000 (Thermo Fisher Scientific, cat. L3000008) and following the manufacturer's instructions. After 48 h, single ESCs expressing high levels of Cas9/gRNAs were selected by FACS sorting (using the EGFP/mCherry fluorescent reporter) and plated into wells of a 96-well plate coated with gelatin and containing serum/LIF medium supplemented with antibiotics (Penicillin-Streptomycin) to avoid contamination. ESC clones were expanded in 24-well plates, and genomic DNA was extracted (Qiagen, cat. 69506) for genotyping *Tet* alleles. PCR genotyping was performed using primer pairs specifically amplifying wild-type or knockout *Tet* alleles. Correctly targeted ESC clones were expanded, and frozen aliquots were transferred to liquid nitrogen tanks for long-term storage. For self-renewal and differentiation assays, a single clone was used for each genotype (T1KO C3, T2KO C26, T3KO C3, DKO C13 and TKO C15).

**List of gRNAs used for generating *Tet* knockout alleles**

| Target locus | gRNA site (excluding PAM sequence) |
| --- | --- |
| *Tet1* start codon | TGCGGGGCGGGACCGAGACA |
| *Tet1* stop codon | TGCGGGACCCTACAATCGTT |
| *Tet2* start codon | GGACAGAACCACCCATGCTG |
| *Tet2* stop codon | GCCATTAGGCCAGACCACCA |
| *Tet3* start codon | GAGTCCATCTGACCTGGAAC |
| *Tet3* stop codon | GGCCCGACGCTGAGGTACGC |

**List of primers used for genotyping *Tet* knockout alleles**

| Target locus | Primer sequence |
|---|---|
| Tet1 FW1 | ACTCCGATGATCCTGCCTCT |
| Tet1 RV1 | TCGGGGTTTTGTCTTCCGTT |
| Tet1 RV2 | CAGCTCATCACTCCGTGTGT |
| Tet2 FW1 | CAGATGCTTAGGCCAATCAAG |
| Tet2 FW2 | AGGATCACACAGGAAGCAGC |
| Tet2 RV1 | AGAAGCAACACACATGAAGATG |
| Tet2 RV2 | CTGTGTCCCACGGTTACACA |
| Tet3 FW1 | CCACCTCTGAGCGCAGAGTG |
| Tet3 FW2 | GATCTGAGCTCTCACAGGGC |
| Tet3 RV1 | GATGAACACAGTTCCTGACAG |
| Tet3 RV2 | CTGCCCACCTGTAACCCATT |

## 5mC and 5hmC analysis

Genomic DNA from wild-type (-cytokines) and $Tet1^{-/-}$, $Tet2^{-/-}$, $Tet3^{-/-}$ TKO (+/− cytokines) day 6 PGCLCs was isolated using the Qiagen Blood and Tissue kit (Qiagen, cat. 69504) following the manufacturer's instructions. Wild-type cells differentiated in the presence of cytokines were sorted for the co-expression of CD61 and SSEA1 (PGCLCs), and genomic DNA was extracted with phenol-chloroform followed by ethanol precipitation. Briefly, sorted cells were lysed in 200 µl of 1X TE, 5 µl of 20% SDS and incubated at 37 °C for 1 h. About 1 µl of RNAse A (10 mg/ml) was then added to the solution, and RNA was digested for 5 min at room temperature. The solution was then supplemented with 1 µl of Proteinase K (Invitrogen, cat. 100005393) and incubated at 50 °C for 3 h. Genomic DNA was then isolated using 200 µl of phenol-chloroform, washed with the same volume of chloroform, and precipitated overnight with 20 µl of 3 M sodium acetate, 1 µl of glycogen (Merck, cat. 10901393001) and 500 µl of ethanol. 5mC and 5hmC were measured using the 5mC DNA ELISA kit and the Quest 5hmC DNA ELISA kit (Zymo Research, cat D5325 and D5425, respectively) following the manufacturer's instructions.

## Immunofluorescence analysis and microscopy

For brightfield analyses, live cells were imaged using an inverted microscope (MOTIC AE2000) and the software Volocity.

For immunofluorescence analyses, cells were washed with PBS and fixed with 4% PFA for 10 min at room temperature. After fixation, cells were washed with PBS and permeabilised with a solution of PBS containing 0.3% (v/v) Triton X-100 for 10 min at room temperature. Samples were blocked in blocking buffer (PBS supplemented with 0.1% (v/v) Triton X-100, 1% (w/v) BSA and 3% (v/v) serum of the same species as the secondary antibodies were raised in) for 1 h at room temperature. Following blocking, samples were incubated with primary antibodies diluted in blocking buffer overnight at 4 °C. After four washes with PBS containing 0.1% (v/v) Triton X-100, samples were incubated with fluorescently labelled secondary antibodies diluted in blocking buffer for 1 h at room temperature in the dark. Cells were washed four times with PBS

containing 0.1% (v/v) Triton X-100. DNA was stained with 4′,6-diamidino-2-phenylindole (DAPI) for 5 min at room temperature. Cells were washed with PBS for 5 min. Samples were imaged by fluorescence microscopy (Nikon Ti-E). Images were analysed and processed using the software Fiji. To allow comparative analyses between cell lines, samples were imaged, processed and analysed in parallel.

PGCLC aggregates were fixed in 4% PFA (Thermo Fisher, cat. J19943) supplemented with 0.1% Triton X-100 (Sigma, cat. X100) for 30 min and then permeabilised overnight at 4 °C in 0.5% v/v Triton X-100 in PBS. Blocking was also performed overnight at 4 °C in blocking buffer (0.1% v/v Triton X-100 in PBS supplemented with 1% BSA and 3% donkey serum). Primary antibodies were then diluted in blocking buffer and incubated at 4 °C for 72 h. PGCLC aggregates were washed three times for 10 min in 0.1% v/v Triton X-100 in PBS and then incubated overnight at 4 °C in secondary antibodies and DAPI diluted in blocking buffer. PGCLC aggregates were then washed three times for 10 min in 0.1% v/v Triton X-100 in PBS and mounted for imaging. Aggregates were mounted in glass-bottom 96-well square plates (Ibidi, cat. 89626) in 1% low-melting-point UltraPure agarose (Thermo Fisher, cat. 16500500) and cleared for 3 h at room temperature with RapiClear 1.47 (Sunjin Lab, cat. RC147001). PGCLC aggregates were then imaged in the high-content confocal microscope Opera Phenix Plus. Image segmentation and signal quantification were done using the Harmony High-Content Imaging and Analysis Software.

**List of antibodies used in this study**

| Antibody target | Reference | Working dilution (application) |
|---|---|---|
| OCT4 | Santa Cruz, sc-8628 | 1:200 (immunofluorescence) |
| NANOG | eBioscience, cat. 14-5761-80 | 1:500 (immunofluorescence) |
| SOX2 | Abcam, ab92494 | 1:500 (immunofluorescence) |
| OCT6 | Santa Cruz, sc-11661 | 1:200 (immunofluorescence) |
| AP2γ | Santa Cruz, sc12762 | 1:200 (immunofluorescence) |
| TUJ1 | Biolegend, 801202 | 1:500 (immunofluorescence) |
| CD15 (SSEA-1) | Biolegend, 125608 | 1:200 (flow cytometry) |
| CD61 | Biolegend, 104307 | 1:600 (flow cytometry) |

## RT-qPCR analysis

Total RNA was isolated using the RNeasy Plus Mini kit (Qiagen, cat. 74136), following the manufacturer's instructions. The quantity and purity of RNA samples were determined using a micro-volume spectrophotometer (Nanodrop, ND-1000). RNA was reverse-transcribed with SuperScript III (Invitrogen, cat. 18080044) using random hexamer oligonucleotides, following the manufacturer's instructions. Triplicate or duplicate qPCR reactions were set up with the Takyon SYBR MasterMix (Eurogentec, cat. UF-NSMT-B0701) and analysed using the Roche LightCycler 480 machine. For all qPCR primer pairs, standard curves were performed to assess the amplification efficiency and melting curves were generated to verify the production of a single DNA species. Plotting and statistical analysis were performed using the software GraphPad Prism version 10.4.2.

### List of primers used for RT-qPCR

| Primer pairs | Forward primer | Reverse primer |
|---|---|---|
| TBP | GGGGAGCTGTGATGTGAAGT | CCAGGAAATAATTCTGGCTCA |
| Tet1 | TTGAGAAGATAGTGTTCACGG | CACTTCTTCTGATCACCCAC |
| Tet2 | CTCATGGAAGAAAGGTATGGAG | GCTCTTGCCTTCTTTACCAG |
| Tet3 | ACTGTCAAGACAGGCTCAG | ATCTCCATGGTACACTGGC |
| Oct4 | GTTGGAGAAGGTGGAACCAA | CTCCTTCTGCAGGGCTTTC |
| Nanog | AGGATGAAGTGCAAGCGGTG | TGCTGAGCCCTTCTGAATCAG |
| Esrrb | CGATTCATGAAATGCCTCAA | CCTCCTCGAACTCGGTCA |
| Rex1 | CAGCTCCTGCACACAGAAGA | ACTGATCCGCAAACACCTG |
| Prdm14 | CCCTACCTGTGTTCAACCT | GTCTCCAGAGTGGACTCTC |
| Ascl1 | GACTTTGGAAGCAGGATGG | CATCTTAGTGAAGGTGCCC |
| Tuj1 | CCCTTCGATTCCCTGGTCA | ACGGCACCATGTTCACAGC |
| Cdh2 / N-cadherin | GCCATCATCGCTATCCTTCT | CCGTTTCATCCATACCACAAA |
| Sox1 | GTGACATCTGCCCCCATC | GAGGCCAGTCTGGTGTCAG |
| Sox17 | CACAACGCAGAGCTAAGCAA | CGCTTCTCTGCCAAGGTC |
| Gata4 | CCTGTGGCCTCTATCACAAGATG | GCGGCGCTGAGGCTTA |
| Flk1 | CCACAGAACAACTCAGGGCTA | GGGAGCAAAGTCTCTGGAAA |
| Col1a1 | TGTTCAGCTTTGTGGACCTC | TCAAGCATACCTCGGGTTTC |
| Otx2 | GTACCCAGACATCTTCATGAG | TTCTTAAACCATACCTGCACC |
| Oct6 | CGCTGTATGCTGTATTTATCGG | ATACACAGATGCGGCTCTC |
| Dppa3 | ACCAAACTGGGTTCCTCCATA | TCCTAATTCTTCCCGATTTTC |
| Ddx4 | TGTAGCACCAACTCGAGAACTG | ACCAAACTGGGTTCCTCCATA |
| Foxa2 | ACCAAACTGGGTTCCTCCATA | ACCAAACTGGGTTCCTCCATA |
| Prdm1 | CCCCTCATCGGTGAAGTCTA | TGGTGGAACTCCTCTCTGGA |
| Tfap2c | ATCCCTCACCTCTCCTCTCC | CCAGATGCGAGTAATGGTCGG |

### RNA-seq analysis

RNA extraction was performed using the Direct-zol RNA MiniPrep kit (Zymo Research, cat. R2052) following the manufacturer's instructions. RNA quality was assessed using Tapestation 2200 (Agilent) and the ratio of RNA and DNA concentrations were determined using Qubit. The NEBNext rRNA Depletion kit (Human/Mouse/Rat) (cat. E6310) was used with 100 ng of total RNA. rRNA-depleted RNA was then DNase-treated and purified using Agencourt RNAClean XP beads (Beckman Coulter Inc., cat. 66514) and the NEBNext libraries were generated using NEBNext Ultra II Directional RNA library prep kit for Illumina (cat. E7760). Sequencing was then performed on the NextSeq 2000 platform (Illumina Inc., cat. 20038897). FASTQ files were generated in Illumina's BaseSpace, and quality trimming was performed with TrimGalore 0.5.0 and cutadapt 1.16 (Martin, 2011). Reads were aligned to the mouse genome (mm10) using STAR 2.7.1a (Dobin et al, 2013), and count matrices were then generated using subread 1.5.2. Differential expression and principal component (PC) analysis were performed using DESeq2 (Love et al, 2014), setting significance thresholds as an adjusted $p$ value lower than 0.05 and a log2 fold change higher than 2 or lower than −2. GO enrichment was performed using clusterProfiler enrichGO function (Yu, 2024), focusing solely on the ontology "Biological processes", $p$ value and $q$ value cutoffs of 0.05 and 0.10, respectively.

### Analysis of publicly available in vivo datasets

PGC data from (Ishikura et al, 2021) was obtained from GEO as raw files and processed following the process described above, keeping with similar mapping and count matrix generation settings to match data generated in this study.

Chimeric embryos data from (Cheng et al, 2022) was obtained using publicly available code https://tanaylab.github.io/tet-gastrulation/index.html#run-the-notebooks. Notebooks 1. "Import scRNA-seq plates" and 2. "FACS gating of the cell population" were run to download data and sort cells. Embryos were then analysed using Seurat_5.0.0 (Hao et al, 2024), SingleCellExperiment_1.30.1 (Amezquita et al, 2020), scater_1.36.0 (McCarthy et al, 2017), and harmony_1.2.3 (Korsunsky et al, 2018), for normalisation, integration, clustering, and dimensionality reduction.

### Statistical analysis

Statistical analysis of RT-qPCR, flow cytometry, ELISA, immunofluorescence and colony counting data was performed using the software GraphPad Prism version 10.4.2. The calculated mean obtained in Tet mutant samples was compared to the mean in the wild-type (control) sample using either the Student's $t$-test or a one-way ANOVA, using relevant methodology to correct for multiple comparisons (Dunnett, Bonferroni or Tukey, depending on datasets). Data were unpaired and assumed to follow a normal distribution. Means between two groups were considered statistically different when the calculated $p$ value was below 0.05 (95% confidence interval). Individual $p$ values for all statistical comparisons are provided in Table EV1. The strength of statistical significance was defined using the following thresholding system: ns (Not significant): $p \geq 0.05$, * (Significant): $0.01 < p < 0.05$, ** (Very significant) $0.001 < p < 0.01$, *** (Extremely significant) $0.0001 < p < 0.001$, **** (extremely significant) $p < 0.0001$.

## Data availability

Original RNA-seq data were deposited at the NCBI Gene Expression Omnibus (GEO) with accession number GSE273732. Data used for integration with in vivo samples is available through NCBI Gene Expression Omnibus (GEO) with accession numbers GSE94136 and GSE168222. Single-cell data for TET-TKO chimeras is available through NCBI Gene Expression Omnibus (GEO) with accession number GSE205917. All other source data for the main figures is available at https://doi.org/10.5281/zenodo.17037372.

The source data of this paper are collected in the following database record: biostudies:S-SCDT-10_1038-S44318-025-00597-9.

## Peer review information

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

## Acknowledgements

We thank Dirk Schübeler (FMI, Basel) for sharing his TET triple-knockout and control cells. We thank Bertrand Vernay for microscopy support, Claire Cryer and Fiona Rossi for assistance with flow cytometry, Justyna Cholewa-Waclaw for assistance with Opera Phenix imaging, the Genetic Core Sequencing Service (WGH, Edinburgh) for library preparation and sequencing of RNA-seq samples, Andrea Corsinotti for qPCR primers, and Val Wilson, Steve Pollard and Richard Meehan for comments on the manuscript. This work was funded by UK Medical Research Council Grants MR/L018497/1 and MR/T003162/1 to IC. EB was supported by a Marie Sklodowska-Curie fellowship (H2020-MSCAIF-2018/843879). RP was supported by an MRC PhD studentship, SBG by an MRC Precision Medicine PhD studentship and TT by a Darwin Trust of Edinburgh PhD studentship.

## Author contributions

**Raphaël Pantier**: Conceptualisation; Data curation; Formal analysis; Investigation; Methodology; Writing—original draft; Writing—review and editing. **Elisa Barbieri**: Formal analysis; Investigation; Writing—original draft;

Writing—review and editing. **Sara Gonzalez Brito**: Formal analysis; Investigation; Writing—original draft; Writing—review and editing. **Ella Thomson**: Formal analysis; Investigation. **Tülin Tatar**: Formal analysis; Investigation. **Douglas Colby**: Formal analysis; Investigation. **Man Zhang**: Formal analysis; Investigation. **Ian Chambers**: Conceptualisation; Supervision; Funding acquisition; Writing—original draft; Writing—review and editing.

Source data underlying figure panels in this paper may have individual authorship assigned. Where available, figure panel/source data authorship is listed in the following database record: biostudies:S-SCDT-10_1038-S44318-025-00597-9.

## Disclosure and competing interests statement

The authors declare no competing interests.

# Expanded View Figures

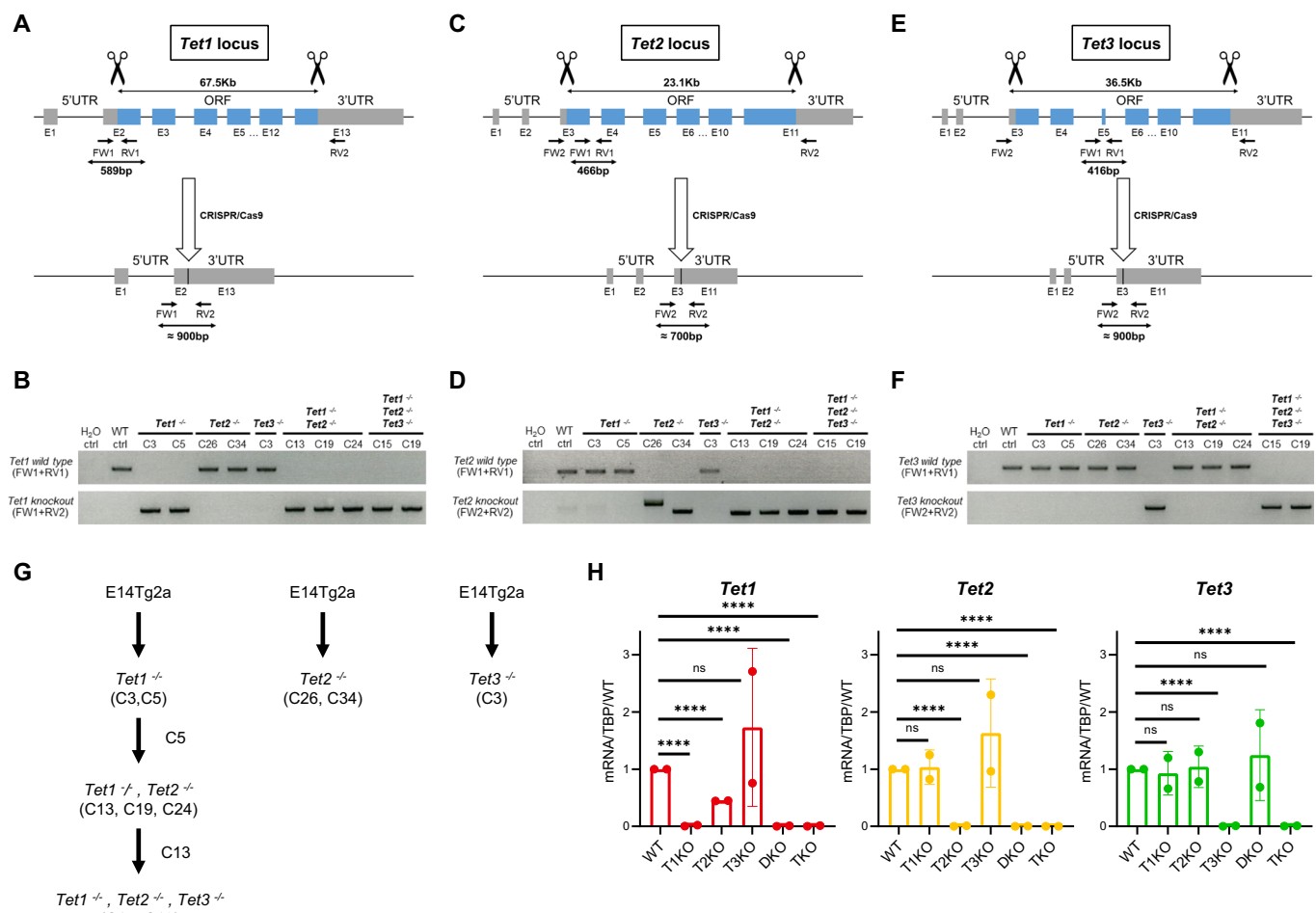

**Figure EV1. Genetic deletion of *Tet1/2/3* open reading frames by CRISPR/Cas9.**

For all loci the general targeting strategy is indicated (**A, C, E**) with two gRNAs (scissors) designed close to the start and stop codons, respectively. The position of genotyping primers and the sizes of PCR products are indicated. Open reading frames are indicated in blue and untranslated regions (UTRs) in grey. (**B, D, F**) Agarose gel analysis of PCR genotyping of *Tet1, 2* and *3* alleles in all cell lines, with the PCR reaction indicated on the left, and the deduced genotype at the top. (**A**) To knockout *Tet1*, wild-type E14Tg2a ESCs were co-transfected with Cas9 and two gRNAs targeting the *Tet1* start and stop codons, respectively. (**B**) Two *Tet1*⁻/⁻ clones carrying deletions of both *Tet1* alleles (C3 and C5) were obtained, as demonstrated by the presence of a PCR product for the knockout allele (FW1 + RV2) and the absence of a PCR product for the wild-type allele (FW1 + RV1). (**C**) *Tet2* knockout and *Tet1, Tet2* double knockout (hereafter referred to as DKO) cell lines were generated from E14Tg2a and *Tet1*⁻/⁻ C5 ESCs using the strategy shown. (**D**) Clones (C26, C34) lacking *Tet2* alleles and clones lacking both *Tet1* and *Tet2* alleles (C13, C19 and C24) were generated. (**E**) *Tet3* knockout and *Tet1, Tet2* and *Tet3* triple knockout (hereafter referred to as TKO) ESCs lacking all six *Tet* alleles were generated from E14Tg2a and *Tet1*⁻/⁻, *Tet2*⁻/⁻ C13 ESCs as illustrated. (**F**) One clone lacking *Tet3* alleles (C3) and two TKO clones lacking all alleles for *Tet1, Tet2* and *Tet3* (C15, C19) were obtained. WT ctrl. Wild-type E14Tg2a ESCs (parental cell line), H₂O ctrl. Control sample with no DNA template. (**G**) Summary of the *Tet* knockout ESC lines generated in this study and their interrelationships (see Fig. 1). Homozygous clone numbers are indicated within brackets. New rounds of targeting are indicated with arrows. (**H**) *Tet1, Tet2* and *Tet3* mRNA levels in the indicated ESC lines. mRNA levels were quantified by RT-qPCR, normalised to TBP mRNA levels, and expressed relative to levels in wild-type E14Tg2a ESCs (mRNA/TBP/WT ESC). Data points: replicate experiments, centre: mean, error bars: standard deviation, n = 2. Stars indicate statistical significance compared to wild-type (Student's t-test). Individual p values are provided in Table EV1.

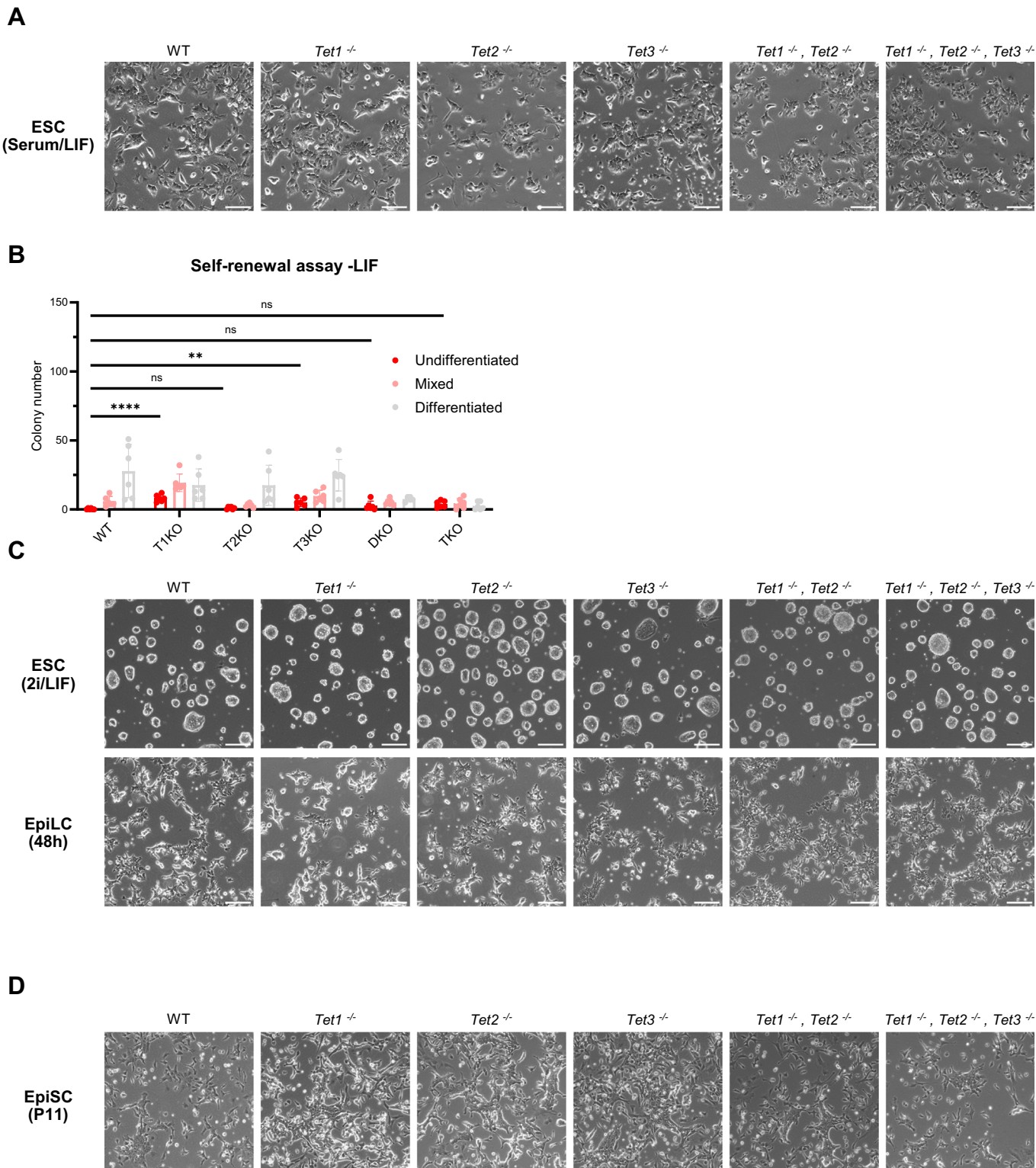

**Figure EV2. Phenotype of TET-deficient lines in different pluripotent states.**

(A) Phase contrast images of the indicated live ESC lines cultured in serum/LIF. Scale bars: 100 μm. (B) Alkaline phosphatase staining of the indicated ESC lines following plating at clonal density in serum-containing medium without LIF for 7 days. Colonies were counted and categorised according to their alkaline phosphatase staining (data points: replicate wells, centre: mean, error bars: standard deviation, $n = 6$). Stars indicate statistical significance compared to wild-type for undifferentiated colonies (one-way ANOVA test). Individual p-values are provided in Table EV1. (C) Phase contrast images of the indicated live ESC lines cultured in 2i/LIF and following 48 h of EpiLC differentiation in activin/FGF. Scale bars:100 μm. (D) Phase contrast images of the indicated live EpiSC lines (passage 11) cultured in activin/FGF. Scale bars:100 μm.

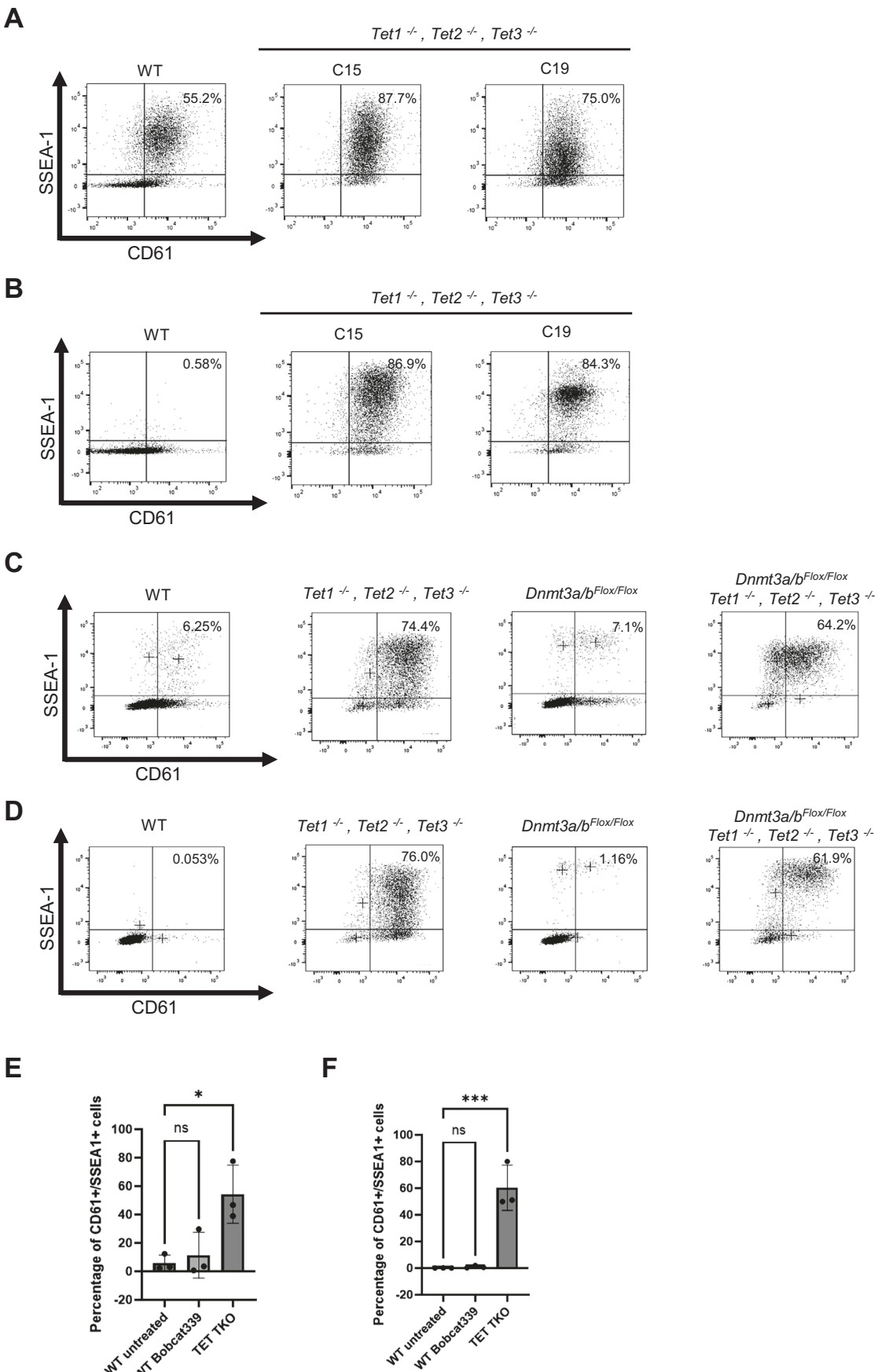

**Figure EV3. Phenotype of independent TET-TKO lines in germline commitment.**

(A, B) Flow cytometric analysis of independent clones (C15 and C19) of $Tet1^{-/-}$, $Tet2^{-/-}$, $Tet3^{-/-}$ (TKO) cell line following 6 days of PGCLC differentiation with (A) or without (B) PGC-promoting cytokines. SSEA-1/CD61 double-positive cells in the top right quadrant report PGCLCs. (C, D) Flow cytometric analysis of indicated cell lines following 6 days of PGCLC differentiation with (C) or without (D) PGC-promoting cytokines. SSEA-1/CD61 double-positive cells in the top right quadrant report PGCLCs. Cell lines used: WT: E14Tg2a, $Tet1^{-/-}$, $Tet2^{-/-}$, $Tet3^{-/-}$: TET triple knockout line generated in this study, $Dnmt3a/b^{Flox/Flox}$: wild-type cell line (Ginno et al, 2020), $Dnmt3a/b^{Flox/Flox}$ $Tet1^{-/-}$, $Tet2^{-/-}$, $Tet3^{-/-}$: TET triple knockout line generated by Ginno et al, 2020. (E, F) Quantification of SSEA-1/CD61 double-positive cells following 6 days of PGCLC differentiation with (E) or without (F) cytokines in the indicated cell line and treatment (data points: three biological replicates, centre: mean, error bars: standard deviation). Stars indicate statistical significance compared to wild-type (one-way ANOVA test). Individual $p$ values are provided in Table EV1.

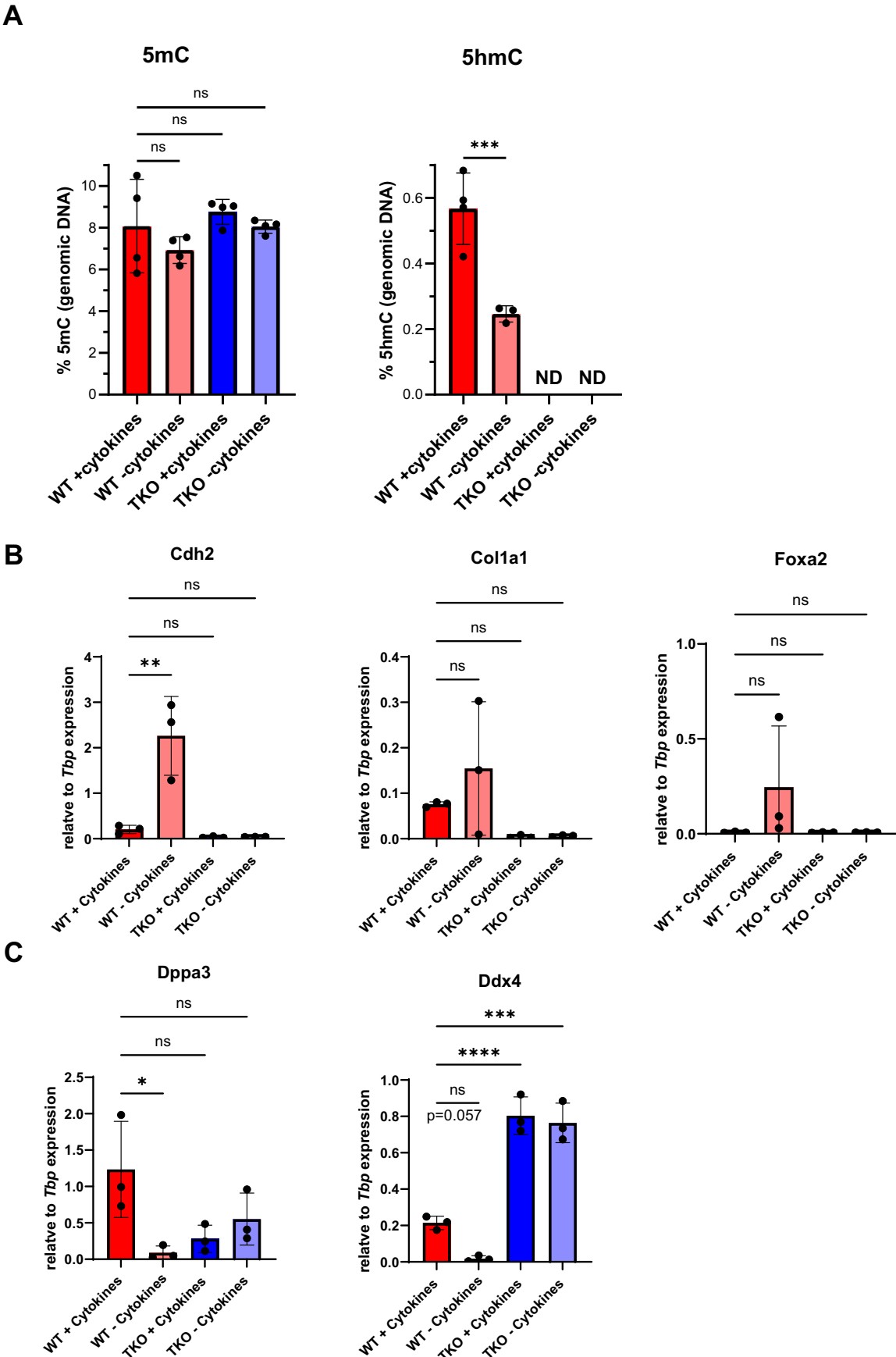

**Figure EV4.  Further characterisation of TET-TKO compared to wild-type PGCLCs.**

(A) 5mC and 5hmC levels measured by ELISA in wild-type and TKO cells at day 6 of differentiation ($+/-$ cytokines). Wild-type cells differentiated in the presence of cytokines were sorted into SSEA1$^+$/CD61$^+$ populations to select for PGCLCs. All other populations were collected as full aggregates (data points: technical and biological replicates, centre: mean, error bars: standard deviation, $n = 2$). Stars indicate statistical significance compared to wild-type (one-way ANOVA test). Individual $p$ values are provided in Table EV1. (B, C) mRNA levels of neural marker *Cdh2*, mesodermal marker *Col1a1*, endoderm marker *Foxa2* (B) and the early germline marker *Dppa3* and the late marker *Ddx4*. (C) RNA isolated from wild-type cells differentiated in the presence of cytokines and TET-TKO cells differentiated either in the presence or absence of cytokines sorted into SSEA1$^+$/CD61$^+$ populations to select for PGCLCs. Wild-type cells differentiated in the absence of cytokines were collected as full aggregates. mRNA levels were quantified by RT-qPCR and normalised to TBP mRNA levels (data points: three biological replicates, centre: mean, error bars: standard deviation, $n = 3$). Stars indicate statistical significance compared to wild-type $+$ cytokines (Student's $t$-test). Individual $p$ values are provided in Table EV1.

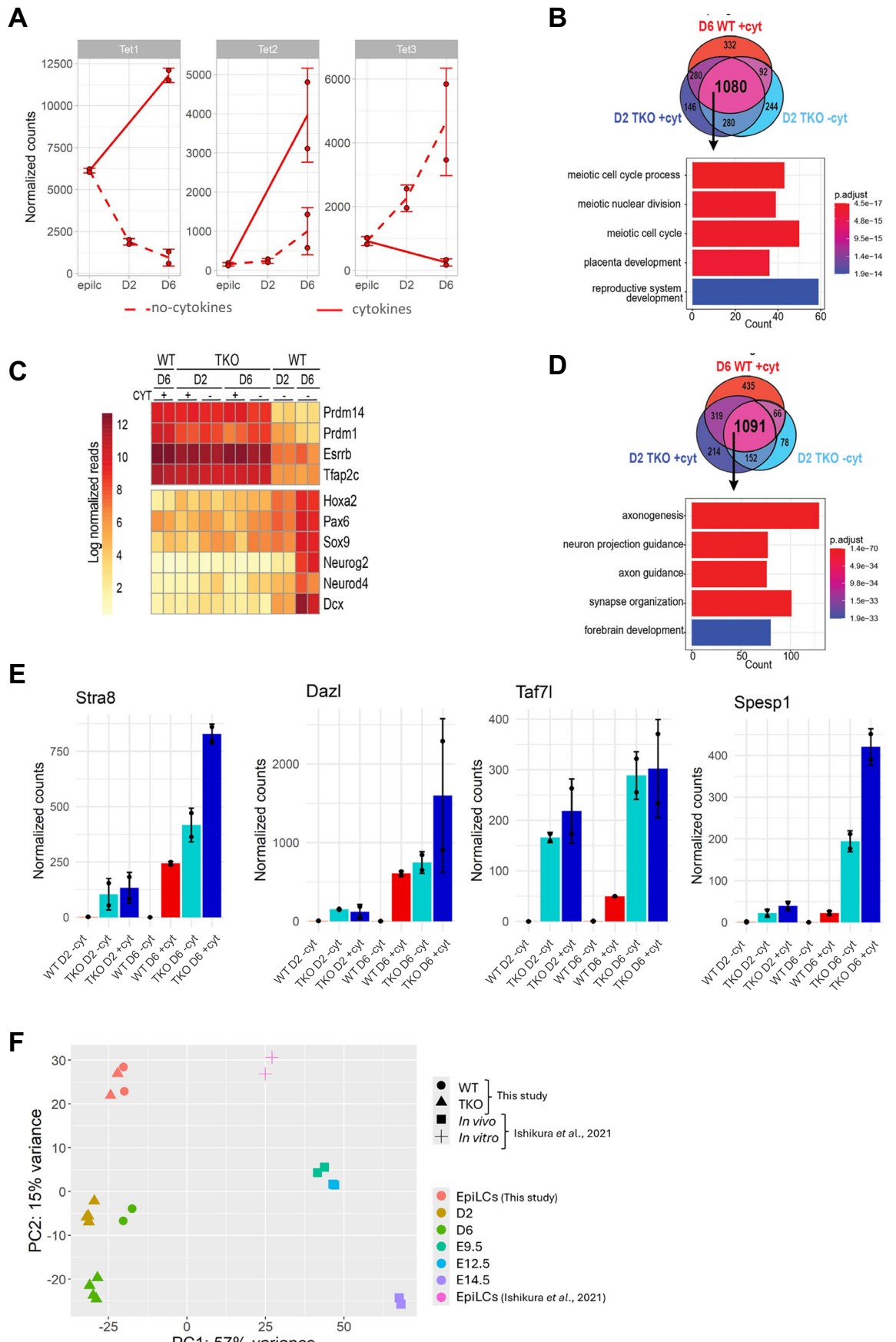

◀

**Figure EV5. Timecourse RNA-seq analysis during TET triple knockout PGCLC differentiation.**

(A) Expression dynamics of *Tet1, Tet2* and *Tet3* in wild-type cells during PGCLC differentiation in the presence and absence of cytokines (RNA-seq reads normalised by DESeq2). Data points: biological replicates, line: mean, error bars: standard deviation, $n = 2$. (B) Overlap of all genes detected as significantly upregulated in day 6 wild-type PGCLCs and day 2 *Tet1*$^{-/-}$, *Tet2*$^{-/-}$, *Tet3*$^{-/-}$ TKO (+/− cytokines) compared to day 6 wild-type cells differentiated without cytokines (*q* value <0.05, Log2 fold change >2). Gene ontology enrichment results are shown for genes shared across all three samples. (C) Log-normalised reads of germline and somatic markers. (D) Overlap of all genes detected as significantly downregulated in day 6 wild-type PGCLCs and day 2 *Tet1*$^{-/-}$, *Tet2*$^{-/-}$, *Tet3*$^{-/-}$ TKO (+/− cytokines) compared to day 6 wild-type cells differentiated without cytokines (*q* value <0.05, Log2 fold change <−2). Gene ontology enrichment results are shown for genes shared across all three samples.
(E) Expression of late germline markers in wild-type and TKO cells during PGCLC differentiation in the presence and absence of cytokines (RNA-seq reads normalised by DESeq2). Data points: biological replicates, centre: mean, error bars: standard deviation, $n = 2$. (F) Principal component analysis (PCA) of bulk RNA-seq samples from this study: Wild-type and TKO cells at day 0 (EpiLCs), day 2 and day 6 of differentiation (+/− cytokines) ($n = 2$); and data from (Ishikura et al, 2021): EpiLCs and PGCs from days E9.5, E12.5 and E14.5 ($n = 2$).

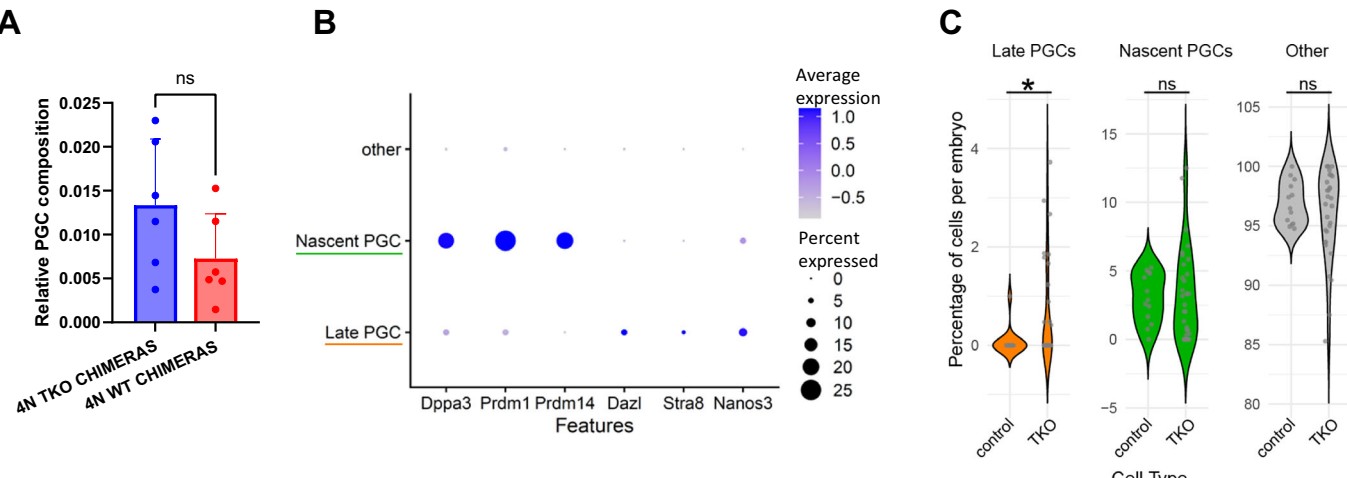

**Figure EV6.  Single-cell analysis of PGCs in wild-type and TET-TKO chimeric embryos.**

(**A**) Relative PGC composition for whole-embryo chimera with transcriptional age >7.75, data from Table S1 in (Cheng et al, 2022). Data points: individual embryos, centre: mean, error bar: standard deviation, $n = 6$. The difference between genotypes was not statistically significant using the Student's $t$-test (ns indicates $p$ value >0.05). (**B**) Expression of germline markers in two newly identified cell clusters: "nascent PGCs" and "late PGCs", all other cells within whole-embryo chimeras classified as "other". Clustering was performed using both wild-type and TKO cells. Colour intensity indicates level of expression, circumference size indicates the percentage of cells within the cluster that express each gene. (**C**) Percentage of each cell type per embryo (Nascent PGCs, Late PGCs and Other). The total number of cells were matched between WT and TET-TKO genotypes (4376 cells for both). Data points: individual embryos, number of embryos WT = 13 TKO = 30. Significance calculated by Student's $t$-test, ns: $p$ value >0.05, Star (*): $p$ value <0.05. Individual $p$ values are provided in Table EV1.

