## [Peer Review File · The EMBO Journal]

TET knockout embryonic stem cells transit efficiently between pluripotent states and exhibit precocious germline entry

Raphaël Pantier, Elisa Barbieri, Sara Gonzalez Brito, Ella Thomson, Tülin Tatar, Douglas Colby, Man Zhang, and Ian

Chambers *Corresponding author: Ian Chambers (i.chambers@ed.ac.uk)*

Review Timeline:

Submission Date:	2nd Dec 24
Editorial Decision:	17th Jan 25
Revision Received:	27th Jun 25
Editorial Decision:	25th Jul 25
Revision Received:	3rd Sep 25
Accepted:	22nd Sep 25

Editor: Daniel Klimmeck

Transaction Report:

Dear Ian,

Thank you again for the submission of your manuscript (EMBOJ-2024-119734) to The EMBO Journal, as well as for your patience with our feedback at this time. As mentioned earlier, your study was assessed by two reviewers with expertise in stem cell biology, developmental gene expression control and epigenetics, whose comments are enclosed below.

As you will see from the experts' reports, the referees acknowledge the analysis and potential interest and value of your findings. However, they also express important issues regarding the completeness of the characterisation of the primordial germ cell-like state induced in the TET mutants (Ref#1, pts.3,4; ref#2, pts.3-5) and consistently point to insufficient mechanistic detail provided (Ref#1, standfirst, pt.5; Ref#2, standfirst, pt.6), which dampen their enthusiasm for the work. Further, the reviewers raise a number of issues related to the presentation of the findings, additional controls and improved methods annotation required, statistics applied and overall discussion of related literature, that would need to be conclusively addressed to achieve the level of robustness and clarity needed for The EMBO Journal.

Given the overall interest stated and broader angle of your findings, we are able to invite you to revise your manuscript experimentally to address the referees' comments. However, please note that the extent of revisions requested appear threshold in our view for the amount of complementary work we typically invite for our venue; also, I need to stress that we do require strong support from the referees on a revised version of the study in order to move on to publication of the work.

I would appreciate if you could contact me during the next weeks for exchange e.g. a video call to discuss your perspective on the comments and potential plan for revisions.

Please feel free to contact me if you have any questions or need further input on the referee comments.

As you know, we generally allow three months as standard revision time. As a matter of policy, competing manuscripts published during this period will not negatively impact on our assessment of the conceptual advance presented by your study. However, we request that you contact the editor as soon as possible upon publication of any related work, to discuss how to proceed. Should you foresee a problem in meeting this three-month deadline, please let us know in advance and we may be able to grant an extension.

When submitting your revised manuscript, please carefully review the instructions below.

Please feel free to approach me any time should you have additional questions related to this.

Thank you for the opportunity to consider your work for publication.

I look forward to your revision.

Best regards,

Daniel

Daniel Klimmeck, PhD
Senior Editor
The EMBO Journal

Instruction for the preparation of your revised manuscript:

- 1) a .docx formatted version of the manuscript text (including legends for main figures, EV figures and tables). Please make sure that the changes are highlighted to be clearly visible.
- 2) individual production quality figure files as .eps, .tif, .jpg (one file per figure).
- 3) a .docx formatted letter INCLUDING the reviewers' reports and your detailed point-by-point response to their comments. As part of the EMBO Press transparent editorial process, the point-by-point response is part of the Review Process File (RPF), which will be published alongside your paper.
- 4) a complete author checklist, which you can download from our author guidelines (<https://wol-prod-cdn.literatumonline.com/pb->

assets/embo-site/Author Checklist%20-%20EMBO%20J-1561436015657.xlsx). Please insert information in the checklist that is also reflected in the manuscript. The completed author checklist will also be part of the RPF.

6) It is mandatory to include a 'Data Availability' section after the Materials and Methods. Before submitting your revision, primary datasets produced in this study need to be deposited in an appropriate public database, and the accession numbers and database listed under 'Data Availability'. Please remember to provide a reviewer password if the datasets are not yet public (see <https://www.embopress.org/page/journal/14602075/authorguide#datadeposition>).

7) Our journal encourages inclusion of *data citations in the reference list* to directly cite datasets that were re-used and obtained from public databases. Data citations in the article text are distinct from normal bibliographical citations and should directly link to the database records from which the data can be accessed. In the main text, data citations are formatted as follows: "Data ref: Smith et al, 2001" or "Data ref: NCBI Sequence Read Archive PRJNA342805, 2017". In the Reference list, data citations must be labeled with "[DATASET]". A data reference must provide the database name, accession number/identifiers and a resolvable link to the landing page from which the data can be accessed at the end of the reference. Further instructions are available at .

8) At EMBO Press we ask authors to provide source data for the main and EV figures. Our source data coordinator will contact you to discuss which figure panels we would need source data for and will also provide you with helpful tips on how to upload and organize the files.

Numerical data can be provided as individual .xls or .csv files (including a tab describing the data). For 'blots' or microscopy, uncropped images should be submitted (using a zip archive or a single pdf per main figure if multiple images need to be supplied for one panel). Additional information on source data and instruction on how to label the files are available at .

9) We replaced Supplementary Information with Expanded View (EV) Figures and Tables that are collapsible/expandable online (see examples in <https://www.embopress.org/doi/10.15252/emj.201695874>). A maximum of 5 EV Figures can be typeset. EV Figures should be cited as 'Figure EV1, Figure EV2' etc. in the text and their respective legends should be included in the main text after the legends of regular figures.

11) For data quantification: please specify the name of the statistical test used to generate error bars and P values, the number (n) of independent experiments (specify technical or biological replicates) underlying each data point and the test used to calculate p-values in each figure legend. The figure legends should contain a basic description of n, P and the test applied. Graphs must include a description of the bars and the error bars (s.d., s.e.m.).

The revision must be submitted online within 90 days; please click on the link below to submit the revision online before 17th Apr 2025.

Referee #1:

The manuscript entitled "TET knockout cells transit between pluripotent states and exhibit precocious germline entry" by Pantier et al. investigates the function of the mouse TET dioxygenase genes in embryonic stem cells. The authors engineer deletions into each of the TET1, TET2, and TET3 genes that remove all coding regions establishing clear loss of function situations. ESC lines with combined deletions of Tet1 and Tet2 as well as all three genes are obtained. Although none of the genes are required for maintenance of ESCs, mutations of Tet1 and combined mutations cause impairments of differentiation. The authors investigate the differentiation defect and show that Tet1/2 and triple mutant (tKO) ESCs form somatic lineages inefficiently, but have increased expression of markers associated with germline development. Different observations are made that lead to the conclusion of a fast entry into germline differentiation. In contrast no effect of forming cell cultures of different pluripotent states including primed pluripotency are observed. The likely most important observation is that Tet1/2 double and tKO cells can form primordial germ cell like cells with high efficiency even when factors are withheld that are required to induce PGCLCs from WT ESCs. This is a noteworthy result that will be of interest for researchers in germline development and pluripotent cell biology. One limitation is that the mechanism behind Tet function in blocking germcell fate and facilitating somatic differentiation remains unclear. The discussion considers that increased DNA methylation or a thus far unknown function of 5hmC might exert differential effect on lineage programs but this is left for future investigation.

Specific points

1. The study makes an unanticipated discovery that loss of Tet activity provides with a very efficient system for germ cell differentiation. Although, the authors show this very elegantly by genetic mutations, the discovery would be practically better exploited by reversible Tet inhibition. Would the authors obtain similar effects on ESC differentiation with a chemical inhibition of Tet enzymes? If this were the case the impact of the manuscript might be substantially increased as genetic ablation of Tet would not be conducive for obtaining late stage gametes such as oocytes.
2. From the text it is not clear how many independent ESC clones have been investigated for each genotype. It would be important to replicate the results in at least two independently established cell lines or if this were impractical rescue the Tet1 mutation of the tKO by transgenic means to guard against unknown clonal effects.
3. In Fig.3 the authors show that CD61 (Integrin beta 1) and SSEA1 double positive PGCLCs are obtained with near 80% efficiency. However, other germ cell markers are not examined. It would be important to perform RT-PCR for Stella (Dppa3) on the sorted PGCLC population and show along the flow cytometry in Fig. 3. Although the authors deduce from their RNAseq that there is no dysregulation of somatic genes, it would be an opportunity to show that in the sorted PGCLCs from tKO EpiLCs without cytokines select neural, mesodermal, and endodermal markers are not simultaneously expressed. This would to my mind be a convincing step to rule out that the mutations leads to misexpression of genes which might include germcell genes.
4. One of the main questions is if the germline obtained in Tet mutant cells is functionally competent to enter gametogenesis. Could the ESCs be injected and PGC development in the embryonic gonad demonstrated? Alternatively, culture systems for meiotic entry might be useful. The authors show by analysis of select markers and transcriptome analysis that the cells are resembling germ cells. However, a biological functional assay would provide a clearer indication.
5. I am puzzled by a potential mechanism for Tet activity underlying both the increased propensity of germline and block in somatic differentiation. Considering that DNA methylation might accumulate in the absence of Tet activity and potentially DNMTs are more active in the soma, could this be tested? Maybe genes are shut down in differentiation by excessive 5mC. Effects on Polycomb might be easily explained secondary effects. Considering the ease of measuring DNA methylation and a proximal mechanistic marker I would encourage the authors to think about including such an experiment.

Minor points

- a) The effect on self renewal is very interesting. On page 7 the authors state (last sentence of 1st paragraph) that Tet activity limits ESC self renewal. This seems in contrast that ground state ESCs have very low levels of DNA methylation and Tet enzymes would likely affect DNA methylation negatively. From this view a positive effect of Tet activity on ESC self renewal

would be expected, which is opposite to the actual observations. It would be interesting to have the authors thoughts how this could be explained and how increased sensitivity to LIF could arise. Is this independent or linked to the propensity of germline differentiation?

Referee #2:

In this study, Pantier et al. study the effects of individual and combined knockouts of the DNA-demethylases Tet1, 2 and 3 on mouse embryonic stem cell pluripotency and differentiation, with particular focus on the germ cell lineage. The study shows that Tet KOs lead to somatic differentiation defects but do not affect transition between naïve, formative and primed pluripotency. The most novel finding from this study however is that Tet KOs lead to enhanced and accelerated differentiation into primordial germ cell like cells (PGCLCs) even in the absence of germ cell-inducing cytokines, suggesting that TET enzymes inhibit germ cell differentiation.

The main findings of this study are potentially of high interest, and the manuscript is clearly written and easy to follow. Nevertheless, the characterization of pluripotent states and PGCLCs is relatively superficial at this point and in the absence of single cell RNA-Seq or DNA-methylation analysis the mechanistic insight remains limited. Adding some of the additional analysis outlined below could therefore enhance the impact of this study substantially and elevate it to the standard of previously published EMBO papers on the germ cell topic.

Major points:

1.) Expanded view figures 2-3 showing differentiation defects of Tet double- and triple-mutant ESCs should be shifted to a main figure as this covers a whole section of the paper. Since there are only 5 main figures, this should not be an issue and will make it easier for the readers to follow.

2.) Lack of statistics: Throughout the manuscript figures featuring bar charts are missing statistics leaving the statements about the phenotypes in the text sometimes subjective. Adding statistics and showing individual data points for each replicate superimposed on the bar charts would solidify the quantitative statements of the paper.

3.) The characterization of PGCLCs is superficial at this point . The RNA-Seq analysis in Fig. 4 is missing ESCs and day 2 wt PGCLCs +cytokines in order to be complete. In the PCA analysis in Fig. 4A also a version without the wildtype cells minus cytokines should be shown, as PC1 seems to be only showing variation for those cells from the other populations. It thereby could mask transcriptional differences between wt and Tet triple KO PGCLCs. Furthermore the inclusion of ESCs would show, if Tet TKO PGCLCs are more similar to wt PGCLCs or revert towards a more pluripotent phenotype. Furthermore, a comparison of the PGCLCs to published RNA-Seq data of in vivo PGCs should be shown to stage the Tet TKO PGCLCs according to their in vivo counterparts. This is in particular relevant for D6 Tet TKO PGCLCs, to assess, if they indeed represent a more advanced germ cell phenotype.

4.) In addition to RNA-data, immunostainings for PGCLC markers such as BLIMP1, TFAP2C or PRDM14 should be provided to give evidence for faithful of PGCLC differentiation of Tet TKO cells.

5.) The wide spread of PGCLC-surface marker expression seen in the FACS plots in Fig. 3A+B suggests a strong heterogeneity in the generated Tet DKO and TKO PGCLC populations. A single cell RNA-Seq analysis would enhance the understanding, if different subpopulation of PGCLCs emerge and a comparison to in vivo PGC datasets would contribute to the understanding of the phenotype.

6.) As TET enzymes catalyze hydroxymethylation as a first step of DNA-demethylation the lack of any type of DNA-methylation and hydroxymethylation analysis is a major limitation of the study. Methylation profiling of the EpiLCs and PGCLCs would provide further mechanistic insight, how Tet enzymes block germline entry.

Minor points:

1. Abstract, line 7: "These TET-deficient ESCs exhibit differentiate defects;..." It should say differentiation defects instead.

2. Page 6, last sentence: "Together, these results extend previous analysis..."
Add references.

Referee #1:

The manuscript entitled "TET knockout cells transit between pluripotent states and exhibit precocious germline entry" by Pantier et al. investigates the function of the mouse TET dioxygenase genes in embryonic stem cells. The authors engineer deletions into each of the TET1, TET2, and TET3 genes that remove all coding regions establishing clear loss of function situations. ESC lines with combined deletions of Tet1 and Tet2 as well as all three genes are obtained. Although none of the genes are required for maintenance of ESCs, mutations of Tet1 and combined mutations cause impairments of differentiation. The authors investigate the differentiation defect and show that Tet1/2 and triple mutant (tKO) ESCs form somatic lineages inefficiently, but have increased expression of markers associated with germline development. Different observations are made that lead to the conclusion of a fast entry into germline differentiation. In contrast no effect of forming cell cultures of different pluripotent states including primed pluripotency are observed. The likely most important observation is that Tet1/2 double and tKO cells can form primordial germ cell like cells with high efficiency even when factors are withheld that are required to induce PGCLCs from WT ESCs. This is a noteworthy result that will be of interest for researchers in germline development and pluripotent cell biology. One limitation is that the mechanism behind Tet function in blocking germcell fate and facilitating somatic differentiation remains unclear. The discussion considers that increased DNA methylation or a thus far unknown function of 5hmC might exert differential effect on lineage programs but this is left for future investigation.

We are pleased that this reviewer considers our results to be "noteworthy" and "of interest for researchers in germline development and pluripotent cell biology". We have responded to the points they raised as detailed below.

Specific points

1. The study makes an unanticipated discovery that loss of Tet activity provides with a very efficient system for germ cell differentiation. Although, the authors show this very elegantly by genetic mutations, the discovery would be practically better exploited by reversible Tet inhibition. Would the authors obtain similar effects on ESC differentiation with a chemical inhibition of Tet enzymes? If this were the case the impact of the manuscript might be substantially increased as genetic ablation of Tet would not be conducive for obtaining late stage gametes such as oocytes.

We agree that testing the effect of Tet inhibitors on PGCLC differentiation would be a fine complementary approach. However, we are concerned that available chemical inhibitors lack the specificity required to give unambiguous results. 2-hydroxyglutarate is the most commonly used TET inhibitor, but it also affects a broad spectrum of 2-oxoglutarate dependent enzymes including histone de-methylases (KDM2A, KDM4A, KDM4C), the HIF prolyl hydroxylase, and even ATP synthase (Xu et al, 2011; Chowdhury et al, 2011; Fu et al, 2015).

"Bobcat339" was developed as an alternative TET inhibitor (Chua et al, 2019), although the biological activity of this molecule has been recently challenged using enzymatic assays and multiple chemical batches (Weirath et al, 2022). Despite this potential caveat, we have tested this chemical inhibitor during PGCLC differentiation. Wild-type EpiLCs were treated with 100 μ M 2H

2. From the text it is not clear how many independent ESC clones have been investigated for each genotype. It would be important to replicate the results in at least two independently established cell lines or if this were impractical rescue the Tet1 mutation of the tKO by transgenic means to guard against unknown clonal effects.

We performed embryoid body, neural monolayer, EpiLC and EpiSC differentiation experiments using single clones for each genotype (T1KO C3, T2KO C26, T3KO C3, DKO C13 and TKO C15, see Figure EV1G). Our results are consistent with several published studies on TET-deficient ESCs (Dawlaty et al, 2014; Li et al, 2016; Verma et al, 2018) and show defects in somatic lineage commitment. Also in agreement with our findings, Wolf Reik's laboratory has recently shown that TET triple knockout does not prevent the transition from naïve to primed pluripotent state (EpiLC differentiation) (Parry et al, 2023, bioRxiv).

Our findings regarding germline entry are novel, and we have now confirmed PGCLC differentiation in the absence and presence of cytokines in a second TKO clones (C19, see Fig. EV1G). This data has been added to the manuscript as Fig EV3A, B).

In addition, we performed PGCLC differentiation in a completely independent TET triple knockout cell line obtained from Dirk Schübeler's laboratory (FMI, Basel) (Ginno et al, 2020), obtaining comparable observations. This data was added as supplementary Figure panels (Fig. EV3C, D). These consistent results give us high confidence that increased germline commitment observed in TKO is a consequence of loss of TET enzymes, rather than a clonal effect. We have updated the manuscript text accordingly.

3. In Fig.3 the authors show that CD61 (Integrin beta 1) and SSEA1 double positive PGCLCs are obtained with near 80% efficiency. However, other germ cell markers are not examined. It would be important to perform RT-PCR for Stella (Dppa3) on the sorted PGCLC population and show along the flow cytometry in Fig. 3. Although the authors deduce from their RNAseq that there is no dysregulation of somatic genes, it would be an opportunity to show that in the sorted PGCLCs from tKO EpiLCs without cytokines somatic markers are not simultaneously expressed. This would to my mind be a convincing step to rule out that the mutations leads to misexpression of genes which might include germcell genes.

We thank the reviewers for the insight and the opportunity to show this new data. As requested, we have assessed the expression of germline and somatic markers on wild-type and TET-triple knockout sorted double-positive cells (SSEA1+/CD61+) at day 6. We used a NO-PGCLC population of unsorted day 6 wild-type cells incubated in the absence of cytokines as a control. TET-TKO PGCLCs do not upregulate the neural marker *Cdh2*, the mesoderm marker *Col1a1*, or the endoderm marker *Foxa2*, showing therefore no upregulation of somatic genes in sorted TET-TKO PGCLCs.

The germline markers *Dppa3* and *Mvh* (*Vasa*) are expressed in TET-TKO PGCLCs, confirming the germline identity of the cells shown on the flow cytometry in Fig. 5. Interestingly, we found differences in the levels of expression between wild-type PGCLCs and TET-TKO PGCLCs, with lower levels of *Dppa3* and higher levels of *Mvh*. We hypothesize that the differences in expression levels compared to wild-type PGCLCs may be related to these cells being at slightly different developmental stages. *In vivo* *Dppa3* is known to be expressed in the founding population of PGC but decreases throughout development, being undetectable at E15.5 (Saitou et al, 2002, Sato et al, 2002), while *Mvh* is not expressed in the nascent PGC population and is instead upregulated in PGCs upon colonization of the genital ridges (Toyooka et al, 2000). Lower levels of *Dppa3* may also be explained, in part, by the previously described role of TET1/2 in the activation of *Dppa3* (Mulholland et al, 2020).

4. One of the main questions is if the germline obtained in Tet mutant cells is functionally competent to enter gametogenesis. Could the ESCs be injected and PGC development in the embryonic gonad demonstrated? Alternatively, culture systems for meiotic entry might be useful. The authors show by analysis of select markers and transcriptome analysis that the cells are resembling germ cells. However, a biological functional assay would provide a clearer indication.

Our paper is focused on the role of TET proteins during germline entry, not gametogenesis which is a process remote in time and one that has already been explored in other studies (e.g. Yamaguchi et al, 2012; Dawlaty et al, 2013; SanMiguel et al, 2018; Hill et al, 2018). From Dawlaty et al, 2013, it is notable that the minority of Tet1/2 double knockout mice surviving after birth remain fertile (with small ovaries and reduced fertility for females). Therefore, Tet1/Tet2 DKO cells are functionally competent to enter gametogenesis.

We apologise for the fact that our manuscript was unclear on this point. We have amended the text on this point more explicit (Introduction, paragraph 2).

5. I am puzzled by a potential mechanism for Tet activity underlying both the increased propensity of germline and block in somatic differentiation. Considering that DNA methylation might accumulate in the absence of Tet activity and potentially DNMTs are more active in the soma, could this be tested? Maybe genes are shut down in differentiation by excessive 5mC. Effects on Polycomb might be easily explained secondary effects. Considering the ease of measuring DNA methylation and a proximal mechanistic marker I would encourage the authors to think about including such an experiment.

As described in Parry et al, 2023, the overall level of DNA methylation (5-mC) in WT and TET TKO EpiLCs does not dramatically differ, suggesting that the absence of TET enzymes does not impact the global increase in methylation occurring during the transition from ESCs to EpiLCs. To assess the level of methylation in PGCLCs, we measured 5-mC by ELISA on genomic DNA samples from sorted WT and TET TKO day 6 PGCLCs. As shown below, the level of 5-mC is slightly lower in somatic cells (unsorted WT -cytokines) compared to PGCLCs generated in TET TKO, both in the presence or absence of cytokines. DNA methylation levels in sorted PGCLCs generated from WT EpiLCs show a high variability, but remain within the range of both somatic cells and TKO PGCLCs. While these results do not provide clear mechanistic insight, they allowed us to exclude the possibility that TET proteins strongly enhance global methylation levels during germline entry.

As detailed in our Discussion section, the view that TETs and DNMTs having antagonistic biological activities may be an oversimplification. This is also difficult to reconcile with phenotypes observed

after loss of function of these proteins. Illustrating this point, DNMT triple-knockout cells also show increased propensity for germline differentiation (Schulz et al. 2024), although the exact effects are distinct from TET knockout. Interestingly, Wolf Reik's laboratory recently proposed that both DNMTs & TETs may be required for cyclic DNA (de-)methylation activities as a way of providing an epigenetic mark for epigenetic reader(s) that may influence cell fate decisions (Parry et al, 2021). Such 'futile cycles' may also allow rapid alterations in gene regulatory circuits during cell fate decisions (Samoilov et al, 2005). We have rewritten the Discussion section to make these points explicit (Discussion, paragraph 7).

Minor points

a) The effect on self renewal is very interesting. On page 7 the authors state (last sentence of 1st paragraph) that Tet activity limits ESC self renewal. This seems in contrast that ground state ESCs have very low levels of DNA methylation and Tet enzymes would likely affect DNA methylation negatively. From this view a positive effect of Tet activity on ESC self renewal would be expected, which is opposite to the actual observations. It would be interesting to have the authors thoughts how this could be explained and how increased sensitivity to LIF could arise. Is this independent or linked to the propensity of germline differentiation?

Again, the enhanced ESC self-renewal could be a consequence of the existence of a regulatory cycle of methylation and demethylation in which the cycle of activities functions to facilitate a rapid change in cell state (differentiation). We have expanded our discussion on this in the revised manuscript (Discussion, paragraph 3)

Referee #2:

In this study, Pantier et al. study the effects of individual and combined knockouts of the DNA-demethylases Tet1, 2 and 3 on mouse embryonic stem cell pluripotency and differentiation, with particular focus on the germ cell lineage. The study shows that Tet KOs lead to somatic differentiation defects but do not affect transition between naïve, formative and primed pluripotency. The most novel finding from this study however is that Tet KOs lead to enhanced and accelerated differentiation into primordial germ cell like cells (PGCLCs) even in the absence of germ cell-inducing cytokines, suggesting that TET enzymes inhibit germ cell differentiation.

The main findings of this study are potentially of high interest, and the manuscript is clearly written and easy to follow. Nevertheless, the characterization of pluripotent states and PGCLCs is relatively superficial at this point and in the absence of single cell RNA-Seq or DNA-methylation analysis the mechanistic insight remains limited. Adding some of the additional analysis outlined below could therefore enhance the impact of this study substantially and elevate it to the standard of previously published EMBO papers on the germ cell topic.

We are pleased that this reviewer considers our results "of high interest", and that "the manuscript is clearly written and easy to follow". We have responded to the points they raised as detailed below.

Major points:

1.) Expanded view figures 2-3 showing differentiation defects of Tet double- and triple-mutant ESCs should be shifted to a main figure as this covers a whole section of the paper. Since there are only 5 main figures, this should not be an issue and will make it easier for the readers to follow.

Expanded view figures 2-3 have been moved to the main figures and relabelled as Figures 1 and 2. The call outs for these figures have been edited in the main text, as have the call-outs for subsequent Figures and EV Figures.

2.) Lack of statistics: Throughout the manuscript figures featuring bar charts are missing statistics leaving the statements about the phenotypes in the text sometimes subjective. Adding statistics and showing individual data points for each replicate superimposed on the bar charts would solidify the quantitative statements of the paper.

Statistics were added to each panel, and all plots now show individual data points.

3.) The characterization of PGCLCs is superficial at this point. The RNA-Seq analysis in Fig. 4 is missing ESCs and day 2 wt PGCLCs +cytokines in order to be complete. In the PCA analysis in Fig. 4A also a version without the wildtype cells minus cytokines should be shown, as PC1 seems to be only showing variation for those cells from the other populations. It thereby could mask transcriptional differences between wt and Tet triple KO PGCLCs. Furthermore the inclusion of ESCs would show, if Tet TKO PGCLCs are more similar to wt PGCLCs or revert towards a more pluripotent phenotype. Furthermore, a comparison of the PGCLCs to published RNA-Seq data of in vivo PGCs should be shown to stage the Tet TKO PGCLCs according to their in vivo counterparts. This is in particular relevant for D6 Tet TKO PGCLCs, to assess, if they indeed represent a more advanced germ cell phenotype.

We have now included ESCs in the PCA as requested (see A) and provided the PCA plot without the wild type cells minus cytokines (see B). This shows that TET TKO PGCLCs are more similar to wt PGCLCs than they are to ESCs, indicating that TET TKO cells do not revert to an ESC-like state during PGCLC differentiation. Moreover, the PC plot without the wild type cells minus cytokines shows that the main source of variability (PC1, accounting for 56% of all variance) reflects temporal differences, with cells distributing from EpiLCs to day 6 PGCLCs, and with day 2 TET-TKO PGCLCs and day 6 WT PGCLCs perfectly aligning in PC1. PC2 shows some variability between WT and TET-TKO cells. Further investigation of genes contributing to PC2 revealed already described differences in germline markers (*Dppa3* – qPCR Figure3; *Stra8* and *Dazl* – Figure EV5E). Panel A above has been added to the manuscript as Figure 6A.

The reviewer also asked that we include day2 wt PGCLCs +cytokines to be complete. To be able to include the Day 6 wt cells, we needed to sort these for CD61/SSEA1 as most of the population in wt cells are not differentiating towards PGCLCs. We would be keen to include the requested data as it would make the analysis more complete, as the reviewer suggests. However, CD61/SSEA1 are not expressed by PGCLCs at day 2, so sorting this minor population is not possible.

We have also provided a comparison to the published data from the Saitou lab (Ishikura et al, 2021). This includes EpiLC data. From the comparison of the EpiLC data, the main difference in PC1 comes from technical differences between our labs. In addition, PC2 comparisons suggest that D6 wt and D2 TETTKO cells are similar to embryonic day 12.5 PGCs, whereas D6 TET TKO cells are more similar to the more advanced E14.5 PGCs. This supports the idea that TET TKO cells are advanced in developmental time. This panel has been added to the manuscript as EVFig 5F.

4.) In addition to RNA-data, immunostainings for PGCLC markers such as BLIMP1, TFAP2C or PRDM14 should be provided to give evidence for faithful of PGCLC differentiation of Tet TKO cells.

Immunostaining of aggregates at day 2 of differentiation shows that TET-TKO aggregates, both in the presence and absence of cytokines, contain a significantly higher number of Ap2y positive cells than wild-type aggregates in the presence of cytokines. This suggests, in agreement with the RNA-seq and FACS data, that TET-TKO cells show enhanced entry into the germline. We thank the reviewers for the opportunity to strengthen our claims with an alternative validation method. These data have been added to the manuscript as Fig 5, panel E.

5.) The wide spread of PGCLC-surface marker expression seen in the FACS plots in Fig. 3A+B suggests a strong heterogeneity in the generated Tet DKO and TKO PGCLC populations. A single cell RNA-Seq analysis would enhance the understanding, if different subpopulation of PGCLCs emerge and a comparison to in vivo PGC datasets would contribute to the understanding of the phenotype.

Heterogeneity in CD61/SSEA1 staining is not unique to our mutant cells. FACS plots for CD61/SSEA1 staining usually show a spread across quadrants rather than very defined populations, as observed also on Fig 5A for wild type cells. While scRNA-seq has the capacity to add more information, we think this would be beyond the scope of the present study. However, we agree that a comparison to in vivo PGC data would be interesting. In addition to the comparison with the Ishikura et al 2021 data (Saitou lab) mentioned above (Figure EV5 F), we have re-analysed the scRNA-seq data from Cheng et al 2022.

This latter study reports a comparison between 4n chimaeras made by injecting either WT or TET TKO mutant cells into 4n embryos. Intriguingly, the Figures in this paper suggest that the PGC compartment may be expanded in TET TKO chimaeras. However, this was not the focus of the Cheng et al 2022 study, and this avenue was not pursued. To address the reviewers' comments, we have re-analysed the Cheng et al dataset to determine whether TET-TKO cells can contribute to the germline in vivo.

Cheng et al 2022 mapped their datasets to the published temporal gastrulation atlas (Mittnenzweig et al., 2021). The mapped cell identities were then used to calculate the relative cell type composition in each whole-embryo chimera (Cheng et al 2022 Table S1 - Relative cell type composition for mixed and whole-embryo chimera with transcriptional age > 7.75). This analysis focused on changes in the somatic populations. However, while not directly reported by Cheng et al 2022, this analysis also shows that the contribution of PGCs towards the composition of whole-embryo chimeras is on average slightly higher in TET-TKO mutants, although this difference is not significant (Below A).

*To more fully address this issue, we re-analysed the Cheng et al 2022 dataset focusing exclusively on germline commitment. PGCs were identified through clustering and marker expression (Below B & C). This analysis revealed two clusters expressing germline markers. The Nascent PGC cluster expressed early markers of PGCs, such as *Dppa3*, *Prdm1* and *Prdm14*, while the Late PGC cluster expressed later germline genes (*Nanos3*, *Dazl* and *Stra8*) (Below B). The distribution and number of cells contributing to each of these clusters were comparable for nascent PGCs (139 WT cells and 144 TKO cells). However, numbers were higher in the case of later PGCs (8 WT cells and 27 TET-TKO cells) (Below C). Importantly, the proportion of Late PGCs per embryo is significantly higher in TET-TKO chimeras, while no significant difference was detected in the Nascent PGC population (Below D).*

This analysis complements our in vitro findings by showing that, similarly to what we find in vitro, TET-TKO chimeras fail to contribute to a number of somatic tissues (Cheng et al 2022). Instead TET-TKO cells contribute with the same, if not higher, efficiency to the PGC population as wild-type cells. Moreover, the distribution of these cells within the UMAP is similar between control and TKO cells (Bellow C), suggesting no drastic increase in heterogeneity nor the existence of any aberrant populations. We hope this analysis satisfies the reviewers' request for in vivo comparisons and scRNA-seq. We would be happy to include this re-analysis of the Cheng data in our manuscript.

6.) As TET enzymes catalyze hydroxymethylation as a first step of DNA-demethylation the lack of any type of DNA-methylation and hydroxymethylation analysis is a major limitation of the study. Methylation profiling of the EpiLCs and PGCLCs would provide further mechanistic insight, how Tet enzymes block germline entry.

As described in the answer to reviewer 1, point 5, Parry et al, 2023 did not identify a drastic difference in the general DNA methylation level in WT and TET TKO EpiLCs, suggesting that the absence of TET enzymes has no major impact on global methylation during the transition from ESCs to EpiLCs. We extended this analysis to DNA methylation in PGCLCs and somatic cells generated from WT and TET TKO. Levels of 5-mC measured by ELISA in TKO PGCLCs (with or without cytokines) are similar to WT PGCLCs and slightly lower in somatic cells (unsorted WT-cytokines).

Moreover, we analysed the level of 5-hmC in day 6 PGCLCs and somatic cells. As expected, the levels of 5-hmC is below detection in TET TKO cells differentiated in either presence or absence of cytokines. Interestingly, the level of 5-hmC is higher in sorted WT PGCLCs vs somatic cells (see bar plot below). This suggests that although TET enzymes are active in the early stages of germline specification, they are not essential for the entry into the germline, as cells lacking TET enzymes and showing no 5-hmC are able to express PGC specific markers.

Minor points:

1. Abstract, line 7: "These TET-deficient ESCs exhibit differentiate defects;..." It should say differentiation defects instead.

This has been changed.

2. Page 6, last sentence: "Together, these results extend previous analysis..."
Add references.

We have added relevant references.

References cited in this Rebuttal letter

Cheng S, Mittnenzweig M, Mayshar Y, Lifshitz A, Dunjić M, Rais Y, Ben-Yair R, Gehrs S, Chomsky E, Mukamel Z, et al (2022) The intrinsic and extrinsic effects of TET proteins during gastrulation. *Cell* 185: 3169-3185.e20

Chowdhury R, Yeoh KK, Tian Y, Hillringhaus L, Bagg EA, Rose NR, Leung IKH, Li XS, Woon ECY, Yang M, et al (2011) The oncometabolite 2-hydroxyglutarate inhibits histone lysine demethylases. *EMBO reports* 12: 463–469

Chua GNL, Wassarman KL, Sun H, Alp JA, Jarczyk EI, Kuzio NJ, Bennett MJ, Malachowsky BG, Kruse M & Kennedy AJ (2019) Cytosine-Based TET Enzyme Inhibitors. *ACS Med Chem Lett* 10: 180–185

Dawlaty MM, Breiling A, Le T, Barrasa MI, Raddatz G, Gao Q, Powell BE, Cheng AW, Faull KF, Lyko F, et al (2014) Loss of Tet Enzymes Compromises Proper Differentiation of Embryonic Stem Cells. *Developmental Cell* 29: 102–111

Dawlaty MM, Breiling A, Le T, Raddatz G, Barrasa MI, Cheng AW, Gao Q, Powell BE, Li Z, Xu M, et al (2013) Combined Deficiency of Tet1 and Tet2 Causes Epigenetic Abnormalities but Is Compatible with Postnatal Development. *Developmental Cell* 24: 310–323

Fu X, Chin RM, Vergnes L, Hwang H, Deng G, Xing Y, Pai MY, Li S, Ta L, Fazlollahi F, et al (2015) 2-Hydroxyglutarate Inhibits ATP Synthase and mTOR Signaling. *Cell Metabolism* 22: 508–515

Ginno PA, Gaidatzis D, Feldmann A, Hoerner L, Imanci D, Burger L, Zilbermann F, Peters AHFM, Edenhofer F, Smallwood SA, et al (2020) A genome-scale map of DNA methylation turnover identifies site-specific dependencies of DNMT and TET activity. *Nat Commun* 11: 2680

Hill PWS, Leitch HG, Requena CE, Sun Z, Amouroux R, Roman-Trufero M, Borkowska M, Terragni J, Vaisvila R, Linnett S, et al (2018) Epigenetic reprogramming enables the transition from primordial germ cell to gonocyte. *Nature* 555: 392–396

Ishikura Y, Ohta H, Sato T, Murase Y, Yabuta Y, Kojima Y, Yamashiro C, Nakamura T, Yamamoto T, Ogawa T, et al (2021) In vitro reconstitution of the whole male germ-cell development from mouse pluripotent stem cells. *Cell Stem Cell* 28: 2167-2179.e9

Li X, Yue X, Pastor WA, Lin L, Georges R, Chavez L, Evans SM & Rao A (2016) Tet proteins influence the balance between neuroectodermal and mesodermal fate choice by inhibiting Wnt signaling. *Proceedings of the National Academy of Sciences* 113: E8267–E8276

Mittnenzweig M, Mayshar Y, Cheng S, Ben-Yair R, Hadas R, Rais Y, Chomsky E, Reines N, Uzonyi A, Lumerman L, et al (2021) A single-embryo, single-cell time-resolved model for mouse gastrulation. *Cell* 184: 2825-2842.e22

Mulholland CB, Nishiyama A, Ryan J, Nakamura R, Yiğit M, Glück IM, Trummer C, Qin W, Bartoschek MD, Traube FR, et al (2020) Recent evolution of a TET-controlled and DPPA3/STELLA-driven pathway of passive DNA demethylation in mammals. *Nat Commun* 11: 5972

Parry A, Krueger C, Lohoff T, Wingett S, Schoenfelder S & Reik W (2023) Dynamic DNA methylation turnover at the exit of pluripotency epigenetically primes gene regulatory elements for hematopoietic lineage specification. 2023.01.11.523441 doi:10.1101/2023.01.11.523441 [PREPRINT]

Parry A, Rulands S & Reik W (2021) Active turnover of DNA methylation during cell fate decisions. *Nat Rev Genet* 22: 59–66

Saitou M, Barton SC & Surani MA (2002) A molecular programme for the specification of germ cell fate in mice. *Nature* 418: 293–300

Samoilov M, Plyasunov S & Arkin AP (2005) Stochastic amplification and signaling in enzymatic futile cycles through noise-induced bistability with oscillations. *Proceedings of the National Academy of Sciences* 102: 2310–2315

SanMiguel JM, Abramowitz LK & Bartolomei MS (2018) Imprinted gene dysregulation in a *Tet1* null mouse model is stochastic and variable in the germline and offspring. *Development* 145: dev160622

Sato M, Kimura T, Kurokawa K, Fujita Y, Abe K, Masuhara M, Yasunaga T, Ryo A, Yamamoto M & Nakano T (2002) Identification of PGC7, a new gene expressed specifically in preimplantation embryos and germ cells. *Mechanisms of Development* 113: 91–94

Schulz M, Teissandier A, De La Mata Santaella E, Armand M, Iranzo J, El Marjou F, Gestraud P, Walter M, Kinston S, Göttgens B, et al (2024) DNA methylation restricts coordinated germline and neural fates in embryonic stem cell differentiation. *Nat Struct Mol Biol* 31: 102–114

Toyooka Y, Tsunekawa N, Takahashi Y, Matsui Y, Satoh M & Noce T (2000) Expression and intracellular localization of mouse Vasa-homologue protein during germ cell development. *Mechanisms of Development* 93: 139–149

Verma N, Pan H, Doré LC, Shukla A, Li QV, Pelham-Webb B, Teijeiro V, González F, Krivtsov A, Chang C-J, et al (2018) TET proteins safeguard bivalent promoters from de novo methylation in human embryonic stem cells. *Nat Genet* 50: 83–95

Weirath NA, Hurben AK, Chao C, Pujari SS, Cheng T, Liu S & Tretyakova NY (2022) Small Molecule Inhibitors of TET Dioxygenases: Bobcat339 Activity Is Mediated by Contaminating Copper(II). *ACS Med Chem Lett* 13: 792–798

Xu W, Yang H, Liu Y, Yang Y, Wang P, Kim S-H, Ito S, Yang C, Wang P, Xiao M-T, et al (2011) Oncometabolite 2-Hydroxyglutarate Is a Competitive Inhibitor of α -Ketoglutarate-Dependent Dioxygenases. *Cancer Cell* 19: 17–30

Yamaguchi S, Hong K, Liu R, Shen L, Inoue A, Diep D, Zhang K & Zhang Y (2012) *Tet1* controls meiosis by regulating meiotic gene expression. *Nature* 492: 443–447

Dear Ian and colleagues,

Thank you for submitting your revised manuscript (EMBOJ-2024-119734R) to The EMBO Journal, as well for your patience with our feedback. Your amended study was sent back to the referees for their scientific reassessment, and we have received reports from both of them, which I enclose below. As you will see, the reviewers state that the work has been substantially enhanced by the revisions and they are now broadly in favour of publication, pending minor amendments.

Thus, we are pleased to inform you that your manuscript has been accepted in principle for publication in The EMBO Journal.

Please carefully consider the remaining minor points raised by referee #2 by adjusting the data presentation and discussion of the findings.

Also, we now need you to take care of a number of issues related to formatting and data presentation as detailed below, which should be addressed at re-submission.

Please contact me at any time if you have additional questions related to below points.

As you might remember from previous experience, every paper at the EMBO Journal now includes a 'Synopsis', displayed on the html and freely accessible to all readers. The synopsis includes a 'model' figure as well as 2-5 one-short-sentence bullet points that summarize the article. I would appreciate if you could provide this figure and the bullet points.

Thank you for giving us the chance to consider your manuscript for The EMBO Journal. I look forward to your final revision.

Again, please contact me at any time if you need any help or have further questions.

Best regards,

Daniel

>> Author Contributions: Remove the author contributions information from the manuscript text. Note that CRediT has replaced the traditional author contributions section as of now because it offers a systematic machine-readable author contributions format that allows for more effective research assessment. and use the free text boxes beneath each contributing author's name to add specific details on the author's contribution.

More information is available in our guide to authors.
<https://www.embopress.org/page/journal/14602075/authorguide>

>> Figure callouts: Please ensure that the figures and panels are called out in sequential order. Currently, Fig 5D is missing a callout.

>> Please rename the 'Conflict of interest' section to 'Disclosure and Competing Interests Statement'.

>> Rename the 'Summary' to 'Abstract'; 'Materials and Methods' should be 'Methods'; please correct the heading for the supplementary figure legends, which should be Expanded View Figure Legends, please correct the EV figures listed in the

legends to "Figure EV1" etc.

>> Add a separate 'Statistical Analysis' section to the Methods part, detailing the algorithms and statistical tests applied.

- The legend for Figure EV5F is missing, please add this to the manuscript text.

>> Source data: additional source data should be provided for Figure EV1 B, D and F as the blots are pixelated under filters and the SD is not uploaded Source data should be uploaded as one (zipped) file per figure or uploaded to the external repository.

>> Reagents and Tools table: Please upload as a separate file using the existing template in the Guide For Authors, listing key reagents, experimental models, software and relevant equipment.

>> Data availability section: please remove the referee token for the GEO dataset and make sure that the data are made publicly accessible.

>> Consider additional changes and comments from our production team as indicated below:

- DAS:

1. Please note that the specific URL for GSE273732 dataset is not provided in the data availability statement.

2. Please note that reviewer access code for GSE273732 dataset is not provided in the data availability statement.

>>> reviewer access added AD 7.7.25

- Figure legends:

1. Please define the annotated p values ****/****/**/* as well as provide the exact p-values for the same in the legend of figure 1B, C; 2B-D; 3A, 4A, B; 5C-E; EV1 H, EV2 B, EV4 A-C as appropriate.

2. Please indicate the statistical test used for data analysis in the legends of figures 4A, B

3. Please note that n=2 in figures 4A, B; 5E, EV1 H, EV4 A, EV5 A, E.

4. Please note that the measure of center for the error bars needs to be defined in the legends of figures 1B, C; 2B-D; 3A, 4A, B; 5C, D, E; EV1 H; EV2 B, EV4 A-C.

Referee #1:

The revised version of the manuscript entitled "TET knockout cells transit between pluripotent states and exhibit precocious germline entry" contains additional data and changes to the text they have addressed my earlier concerns in a satisfactory way. In particular, the authors have now added an independent TET mutant ES cell line which confirms enhanced germline differentiation. In addition, the authors have investigated the use of chemical TET inhibition. Although this line of experimentation has not led to additional insight, it is clear that current limitations of TET inhibitors do not provide additional experimental opportunities. From the newly performed characterization of PGCLCs it is reasonable to assume that lack of TET enzymes does not lead to misspecification but primordial germ cells are properly specified. The study will be of high interest for pluripotent stem cell biology and DNA modification research.

Referee #2:

I appreciate the efforts of the authors to address all points of the reviewers in a detailed manner. Some new analysis and

rewriting have been performed solidifying the conclusions of the paper and I think that new data shown in the rebuttal letter should be included in the manuscript (see my comments below). While a more extensive analysis of DNA-methylation and hydroxymethylation might have provided potentially deeper mechanistic insight, the phenotype of enhanced PGCLC differentiation efficiency even in the absence of cytokines in TET-KO cells is in itself very interesting and worth reporting. I therefore would support publication of the manuscript after implementation of the changes below.

1. Referee 1, Major point 1: TET inhibitor experiment (with Bobcat339):

It would be very informative for the reviewers to include this experiment in the paper as a figure or supplementary figure panel. The Figure would need proper labelling of the Y-axis and statistics and a description of the experimental setup (how long the treatment was - I guess full time during EpiLC induction). Ideally TET inhibitor should have been added as well during PGCLC induction.

2. Referee 1, Major Point 5 and Referee 2, Major Point 6: DNA-methylation assessment

Also these figure panels (methylation and hydroxymethylation assessment) should be included in the paper, since this is a key question, how DNA-methylation is affected in the TET-KO cells. It is a bit disappointing that no deeper methylation analysis has been performed (e.g. enzymatic methyl seq) since this would have potentially revealed mechanistic insight, if specific promoters of key differentiation genes might have been differentially methylated in the TET KO cells, potentially controlling their expression.

3. Referee 2, Major Point 5: I appreciate the efforts of the authors of integrating their data with in vivo datasets and of reanalyzing the in vivo data of Cheng et al 2022. I feel that this new analysis should become part of the paper, since it strengthens the author's in vitro observation that TET KO cells show an advanced PGCLC phenotype, but at the same time puts it into perspective, since the in vivo effect from the Cheng et al data regarding enhanced PGC specification is less dramatic than the author's in vitro data. The authors should include this analysis and provide a balanced discussions about the similarities and differences between in vitro and in vivo observations. This would strengthen the value of the paper for the readers substantially.

The authors addressed the remaining editorial issues.

Dear Ian,

Thank you for submitting the revised version of your manuscript. I have now evaluated your amended manuscript and concluded that the remaining minor concerns have been sufficiently addressed.

I am thus pleased to inform you that your manuscript has been accepted for publication in the EMBO Journal.

On a different note, I would like to alert you that EMBO Press offers a format for a video-synopsis of work published with us, which essentially is a short, author-generated film explaining the core findings in hand drawings, and, as we believe, can be very useful to increase visibility of the work. Please see the following link for representative examples and their integration into the article web page:

<https://www.embopress.org/doi/full/10.15252/emj.2019103932>

Best regards,

Daniel

Daniel Klimmeck, PhD
Senior Editor
The EMBO Journal
EMBO
Postfach 1022-40
Meyerohofstrasse 1
D-69117 Heidelberg
contact@embojournal.org
